# DogRot: Taming Highly Ill-Conditioned Sensing Matrix in Sparse Signal Recovery by Random Gaussian Rotator

## Abstract

Recovering sparse signal from an undetermined system, known as compressing sensing (CS), has been a topic with longstanding interests in many signal processing and machine learning applications. A sensing matrix with low inter-column coherence is fundamental to the identifiability of CS. In many real-world problems (e.g., magnetic resonance imaging reconstruction and genetic disease classification) however, relative sensing matrices could be extremely 'fat', and naturally contain many proportional columns. Solving the resultant CS problems is notoriously fragile. This work aims to address a family of CS problems induced by such ill-conditioned sensing matrices. We propose DogRot, a plug-and-play preconditioner constructed from a designated diagonal-dominant Gaussian random rotator. Intuitively, DogRot reshapes the sensing matrix to lower its mutual coherence while preserving the sparse solution set, thereby strengthening identifiability. We rigorously establish these properties in theory and validate them extensively in practice. As a lightweight and easily integrable preconditioner, DogRot can be seamlessly combined with existing sparse recovery algorithms. Across diverse applications, our experiments show that DogRot consistently reduces mutual coherence and effectively improves the quality of sparse signal recovery.

## 1    Introduction

Sparse signal recovery or compressed sensing (CS) (Donoho, 2006), has been a workhorse since the late 2000s in signal processing and machine learning communities. Specifically, given the observation $\boldsymbol{y} \in \mathbb{R}^n$ and a known fat sensing matrix $\boldsymbol{A} \in \mathbb{R}^{p \times n}$, CS seeks to solve the following *undetermined* system (Donoho, 2006; Candes et al., 2006):

$$\boldsymbol{y} = \boldsymbol{A}\boldsymbol{x} + \boldsymbol{n}, \boldsymbol{x} \in \mathbb{R}^n, \boldsymbol{n} \in \mathbb{R}^p, p < n, \tag{1}$$

where $\boldsymbol{x}$ is the targeted *sparse* signal, and $\boldsymbol{n}$ denotes measurement noise. We represent the support set of $\boldsymbol{x}$ as $\mathcal{S}$ and $|\mathcal{S}| \ll n$. Formulation (1) aligns with a plethora of real-world problems, e.g., real-time visual tracking, (Zhang et al., 2012), remote sensing (Herman & Strohmer, 2009), accelerated magnetic resonance imaging (MRI) (Lustig et al., 2007a), and sparse channel estimation (Gao et al., 2015), where the amount of samples is significantly smaller than that of underlying factor.

A fundamental condition underpinning the identifiability of CS is the so-called *restricted isometry property* (RIP) (Candes & Tao, 2005; Candes, 2008; Candes & Wakin, 2008). Intuitively, RIP says that the columns of sensing matrix $\boldsymbol{A}$ should be sufficiently incoherent, under which solving (1) is feasible via iterative optimization (Tibshirani, 2018; Beck & Teboulle, 2009; Boyd et al., 2011), greedy pursuit (Pati et al., 1993; Needell & Vershynin, 2010; Needell & Tropp, 2009) or Bayesian learning (Ji et al., 2008; Baron et al., 2010). However, it is NP-Hard to check the RIP of a certain matrix. Alternatively, an effective surrogate called *mutual coherence* (MC) [1] of $\boldsymbol{A}$ is widely adopted to evaluate the identifiability of (1) (Elad, 2007; Donoho & Elad, 2003). Hence, much of the existing analysis (including our upcoming analysis) adopts MC as the key metric.

While effective CS approaches under ideal RIP or MC are extensive, in this work however, we focus

---

[1] The MC of sensing matrix $\boldsymbol{A}$ is defined as $\mu(\boldsymbol{A}) = \max\limits_{i \neq j} \left[ \widehat{\boldsymbol{A}}^\top \widehat{\boldsymbol{A}} \right]_{i,j}$, where $\widehat{\boldsymbol{A}}$ is the normalized version of $\boldsymbol{A}$, i.e., $\widehat{\boldsymbol{A}}(:, i) = \boldsymbol{A}(:, i)/\|\boldsymbol{A}(:, i)\|_2$.

on a challenging family of (1) where sensing matrix $A$ is *highly ill-conditioned* and accordingly the RIP is severely violated. Such highly ill-conditioned (1) typically arises when $p \ll n$ and the columns of $A$ are strongly correlated or nearly proportional. For example, in *MRI reconstruction* (Lustig et al., 2007b; Eksioglu, 2016; Ryu et al., 2019; Ding et al., 2023), sub-sampling leads to the number of pixels far exceed the acquired k-space samples, while simultaneously causing the sensing basis to exhibit aliasing under common sampling patterns; In *localized statistical channel modeling* (LSCM) (Luo et al., 2023; Zhang et al., 2024; 2025), different beams exhibit strong similarity, and the dimensionality of underlying angular power spectrum (APS) is way larger than that of measured beams; In *gene-based disease classification* (GDC) (Golub et al., 1999; Zou & Hastie, 2005; Li & Li, 2008), there are oftentimes thousands of genes compared with merely tens of patient samples.

We propose a precondition framework to address the highly ill-conditioned version of (1). Our main contribution is introducing a pre-conditioner $Q \in \mathbb{R}^{n \times n}$, which re-understands (1) as

$$y = AQQ^{-1}x + n. \tag{2}$$

We tailor a special structure for $Q$, termed the *diagonal-dominant random Gaussian rotator* (DogRot). From the theory side, we establish that the tailored invertible $Q$ refines $A$ towards higher incoherence, as well as approximately preserves the solution of $x$. As a consequence, the transformed sensing matrix enables more reliable support recovery for $\mathcal{S}$ using standard sparse recovery algorithms, and the associated coefficients $x_{\mathcal{S}}$ can also be accordingly reconstructed. Extensive experimental results verify that the proposed DogRot consistently reduces the MC of sensing matrices across a variety of applications, with reductions from as high as 0.99 (nearly parallel columns) to as low as 0.33 in certain cases. Moreover, such reductions in MC are shown to translate directly into substantial improvements in the reconstruction quality of the signal of interest.

## 2 BACKGROUND: SPARSE SIGNAL RECOVERY UNDER HIGHLY ILL-CONDITIONED SENSING MATRIX

In this section, we first characterize problem (1) under *highly ill-conditioned sensing matrix* (HIM), and then briefly review some of its close relatives.

### 2.1 HIGHLY ILL-CONDITIONED SENSING MATRIX

**Definition 2.1** (**Highly Ill-conditioned Matrix**). We say that sensing matrix $A$ in (1) is highly ill-conditioned, if it exhibits the following tendencies:
• **High Aspect Ratio:** $p \ll n$, i.e., $n/p \geq \alpha$ for some large constant $\alpha$, e.g., $\alpha \approx 10$ in MRI reconstruction and $\alpha \approx 100$ in GDC.
• **Large Size:** $n/1000 > 1$, e.g., $n = 6552$ in (Zhang et al., 2024), $n = 7192$ in (Golub et al., 1999).
• **High MC:** $\mu(A) \approx 1$, i.e., some of the columns in $A$ has strong co-linearity.
• **Heavy-Tailed Correlations:** The off-diagonal entries of $\widehat{A}^\top \widehat{A}$ have a distribution with a heavy tail near 1; that is, for some $\delta > 0$ and $i \neq j$, $\mathbb{P}(|[G_A]_{i,j}| > 1 - \delta)$ is non-negligible.

An illustration for the defined HIM is provided in Figure 3a in the Appendix F. The large problem scale and limited observations of HIM make it difficult for conventional algorithms to distinguish between its resembled columns, e.g., sub-sampled Fourier bases, closely spaced beams, and genes with highly similar expression profiles. Consequently, recovering $x$ from (1) is severely hindered by these HIM properties. In particular, the first two properties confine a large portion of columns to a low-dimensional subspace, resulting in strong inter-column correlations and expanding the feasible search space. Meanwhile, the latter two properties imply that a considerable fraction of columns are nearly indistinguishable in the observation space. In extreme cases, successful recovery is only guaranteed when $x$ is 1-sparse (Elad, 2007), which is clearly too restrictive for most real-world applications. These observations highlight the detrimental impact of HIMs on sparse recovery, i.e., an accurate reconstruction becomes virtually unattainable in their presence.

### 2.2 PRIOR ARTS

Many approaches have been explored to enhance the performance via manipulating $A$ to reduce its $\mu(A)$ (Elad, 2007; Bruckstein et al., 2008; Abolghasemi et al., 2010; Li et al., 2017; Kilic et al.,

2022; Wang et al., 2024). Among them, projection is the most straightforward and thus most commonly adopted precondition strategy, as it preserves the solution set while reshaping the sensing matrix. This is achieved by imposing an invertible projector $P$ on $A$ to minimizes the MC of the effective observation matrix $D = PA$. Relaxation (Abolghasemi et al., 2010) and decomposition-based methods (Elad, 2007; Kilic et al., 2022) have been two main branches to find an effective projector. Relaxation-based methods aim to minimize a smooth surrogate of $\mu(D)$, as it is non-trivial to express $\mu(D)$ as a tractable objective of $D$. Gradient descent (GD) methods is oftentimes adopted to obtain a stationary solution. On the other hand, decomposition-based approaches exploit structured matrix factorizations, e.g., triangular decomposition and singular value decomposition (SVD) (Elad, 2007), to improve the conditioning of $A$ through well-designed transformations (Elad, 2007; Kilic et al., 2022).

Although these approaches have shown theoretical and empirical success in various applications, they offer limited improvement for the considered HIM, as evidenced in Appendix F. The failure stems from the intrinsic characteristics of HIM. First, when $p \ll n$, the projector $P$ contains far fewer degrees of freedom than $A$, and thus cannot sufficiently decorrelate a large set of nearly parallel vectors. Second, mutual coherence is fundamentally governed by inter-column correlations within $A$, which left multiplication by $P$ cannot directly alter. In addition, relaxation-based methods are vulnerable to local optima, while the extreme ill-conditioning of HIM further undermines the stability and effectiveness of decomposition-based strategies. These limitations highlight the need for new preconditioning techniques explicitly designed for HIM.

Although perfect recovery is favorable in general, in those challenging case with HIM, it may be more realistic to lower our expectations. In many real-world applications, reliably identifying the support of $x$ alone already suffices. For example, when applying the generic approach to disease classification based on gene expression (Golub et al., 1999), the primary interest lies in determining which genes are associated with a disease, rather than in precisely quantifying their contribution factors. Motivated by this, our objective in this work is to achieve a more robust estimation of the support set of $x$ under HIM conditions. Accordingly, the remainder of this paper will focus on enhancing support recovery when the sensing matrix exhibits HIM properties.

## 3 REMOULDING HIM COMPRESSED SENSING VIA PRE-CONDITIONING MATRIX

In this section, we introduce a new preconditioning paradigm designed to address HIM-induced instances of (1). The key idea, expressed as (2), is to apply a structured matrix $Q$ that refines the conditioning of $A$ while preserving the essential structure of (1). In what follows, we provide the theoretical foundations and implementation details of this idea.

### 3.1 SOME INSIGHTS OF PRECONDITIONING

By definition, MC is a column-related metric, so directly manipulating the columns is the most effective way to reduce it. The heavy-tailed correlation distribution of HIM indicates that some highly correlated columns in $A$ already contribute to an HIM $A$, while many others may exhibit weak correlations. This observation motivates the idea of *mixing*, i.e., by combining highly correlated columns with other less correlated ones, the MC of $A$ might be effectively reduced. Such a mixing operation can be realized through right-multiplication with an invertible matrix, which we refer to as a *mixer* in this paper. In this way, the original problem (1) is reformulated as (2), where the new sensing matrix $AQ$ is expected to exhibit reduced MC, thereby enabling improved recovery.

To ensure the recoverability of the original sparse signal, $Q$ must be invertible. Furthermore, since the new problem remains under-determined, the new signal $Q^{-1}x$ must preserve the sparsity of the original one. In short, an effective mixer $Q$ should both decorrelate the columns of $A$ and maintain the sparsity structure of the signal through its inverse.

Overall, the problem of finding an appropriate mixer can be formulated as

$$\min_{Q} \quad \mu(AQ) \tag{3}$$

$$\text{s.t.} \quad Q^{-1}Q = QQ^{-1} = I_n, \tag{3a}$$

$$\text{supp}(Q^{-1}x) = \text{supp}(x) = \mathcal{S}, \tag{3b}$$

where $\text{supp}\,(x) = \{i\,|\,x_i \neq 0\,\}$ denotes the support set of $x$. As previously discussed, the objective to minimize the MC is inherently non-convex. Moreover, the constraints that enforce the invertibility of $Q$ and preserve the support set are themselves highly non-convex. Together, these characteristics render the optimization problem intractable for conventional optimization methods.

## 3.2 Diagonal-Dominant Random Gaussian Rotator

An annoying fact is that finding a global or even local optimizer for (3) is highly non-trivial. While it is relatively easy to search for a matrix that reduces $\mu(AQ)$, simultaneously ensuring invertibility and support preservation is far more difficult. To proceed, we instead resort to structuring a family of pre-conditioning matrices $Q$, termed **D**iagonal-**d**ominant **G**aussian **R**andom **Rot**ator (DogRot). The DogRot, although constructed in a heuristically, can later be theoretically shown to meet our desire. The definition of DogRot is given below:

**Definition 3.1** (**Diagonal-Dominant Gaussian Random Rotator**). Given $n \geq 2$ and a small positive parameter $\epsilon > 0$, a DogRot $Q_{\epsilon,n} \in \mathbb{R}^{n \times n}$ is defined by

$$Q_{\epsilon,n} = \begin{bmatrix} 1 & -q_{2,1} & \cdots & -q_{n,1} \\ q_{2,1} & 1 & \cdots & -q_{n,2} \\ \vdots & \vdots & \ddots & \vdots \\ q_{n,1} & q_{n,2} & \cdots & 1 \end{bmatrix} \quad \text{and} \quad q_{i,j} \begin{cases} = 1 & , i = j \\ \sim \mathcal{N}\,(0, \epsilon)\,, i > j \\ = -q_{j,i} & , i < j \end{cases} \tag{4}$$

By definition, DogRot intuitively encompasses three key properties:

• **Diagonal-dominant:** The diagonal entries are set to 1 while the off-diagonal entries are random variables with a small variance. As a result, the diagonal terms dominate in magnitude, ensuring that the matrix remains close to the identity;

• **Gaussian Random:** Each off-diagonal element below the main diagonal is independently drawn from a Gaussian distribution introducing controlled randomness to the matrix;

• **Rotator:** The DogRot is a combination of the identity matrix and a skew-symmetric matrix. This structure aligns with the Lie algebra $\mathfrak{so}(n)$, which consists of real skew-symmetric matrices, infinitesimal generators of the special orthogonal group $\text{SO}(n)$, making it a random rotator. Consequently, right-multiplying by a DogRot can be viewed as applying a small, random rotational perturbation to each column of the sensing matrix.

Given these properties, some identities of DogRot is readily within reach. We will discuss step by step how the proposed DogRot acts as a feasible solution of problem (3).

### 3.2.1 Invertibility

**Lemma 3.1.** *Let $Q_{\epsilon,n}$ be an $n$-dimensional DogRot. Then, $Q_{\epsilon,n}$ is non-singular almost surely.*

*Proof.* We briefly state the proof here as it is quite straightforward. It is observed that $Q_{\epsilon,n}$ can be decomposed as an identity matrix and a skew-symmetric matrix, i.e., $Q_{\epsilon,n} = I_n + \hat{Q}_{\epsilon,n}$, where $\hat{Q}_{\epsilon,n}$ is skew-symmetric. Since all eigenvalues of a real skew-symmetric matrix are purely imaginary or zero, the eigenvalues of $Q_{\epsilon,n}$ are of the form $1 \pm cj$, which are nonzero. Thus, the random matrix $Q_{\epsilon,n}$ is invertible with probability 1. $\qquad\square$

According to Lemma 3.1, the proposed DogRot is guaranteed to satisfy the constraint in (3a). In fact, we can reach a much stronger conclusion for DogRot, which later offers a more handy way to analyze the influence of DogRot, as provided later in Corollary 3.5.

**Proposition 3.2** (**Transpose Invariance**). *Let $Q_{\epsilon,n}$ be an $n$-dimensional DogRot. Its transpose $Q_{\epsilon,n}^T$ is also a DogRot.*

This proposition states that the class of DogRot matrices is closed under matrix transposition. The proof directly follows from the skew-symmetric definition of the DogRot and the symmetry of Gaussian distribution.

**Theorem 3.3** (**Norm Concentration**). *Let $\boldsymbol{Q}_{\epsilon,n}$ be an $n$-dimensional DogRot and $\boldsymbol{q}_i$ be an arbitrary column in $\boldsymbol{Q}_{\epsilon,n}$. Then, for all $\varepsilon \in (0,1)$, the following inequality holds:*

$$P\left\{\left|\frac{\|\boldsymbol{q}_i\|_2^2}{\epsilon(n-1)+1} - 1\right| \geq \varepsilon\right\} \leq 2\exp\left(-\frac{(n-1)\varepsilon^2}{8}\right). \tag{5}$$

The proof is provided in Appendix A. This theorem states that the probability that the norm of an arbitrary column in the DogRot deviates significantly from $\epsilon(n-1)+1$ decays exponentially with $n$. In other words, the norm of each column in $\boldsymbol{Q}_{\epsilon,n}$ is sharply concentrated around $\epsilon(n-1)+1$ with a large $n$.

**Theorem 3.4** (**Asymptotic Orthogonality**). *Let $\boldsymbol{Q}_{\epsilon,n}$ be an $n$-dimensional DogRot. Let $\boldsymbol{q}_i$ and $\boldsymbol{q}_j$ denote two distinct columns. Then, for all $\varepsilon \in (0,1)$, we have*

$$P\left\{\left|\frac{\boldsymbol{q}_i^T\boldsymbol{q}_j}{\epsilon(n-1)+1}\right| \geq \varepsilon\right\} \leq 4\exp\left(-\frac{(n-1)\varepsilon^2}{8}\right). \tag{6}$$

The proof is provided in Appendix B. In other words, $\lim_{n\to\infty} \boldsymbol{q}_i^T\boldsymbol{q}_j = 0$ in probability. Theorem 3.4 implies that any two distinct columns of $\boldsymbol{Q}_{\epsilon,n}$ become asymptotically orthogonal as $n$ grows. From the above two theorems, we can arrive at the following property:

**Corollary 3.5.** *Let $\boldsymbol{Q}_{\epsilon,n}$ be an $n$-dimensional DogRot with a large $n$. Then, $\boldsymbol{Q}_{\epsilon,n}$ is asymptotically a scaled orthogonal matrix, i.e., $\boldsymbol{Q}^T\boldsymbol{Q} \approx \boldsymbol{Q}\boldsymbol{Q}^T \approx (\epsilon(n-1)+1)\boldsymbol{I}_n$.*

The proof is omitted here since Theorem 3.3 and 3.4 directly lead to this conclusion. Corollary 3.5 says that $\boldsymbol{Q}_{\epsilon,n}$ can be regarded as a scaled orthogonal matrix in the asymptotic sense. Recalling that the HIM defined in this work has a large number of columns, i.e., $n$ is sufficiently large, hence, the DogRot matrix used as its mixer satisfies the properties of norm concentration and orthogonality. Consequently, we can use $\boldsymbol{Q}_{\epsilon,n}^T$ to approximate $\boldsymbol{Q}_{\epsilon,n}^{-1}$ in the following analysis. Note that we have shown the transpose invariance of the DogRot, hence $\boldsymbol{Q}_{\epsilon,n}^T$ also possesses the above properties.

### 3.2.2 THE EFFECT OF DOGROT: REDUCING MUTUAL COHERENCE

With the above preparations, we are now ready to demonstrate how $\boldsymbol{Q}_{\epsilon,n}$ reduces the MC of $\boldsymbol{A}$ while preserving the sparsity of $\boldsymbol{x}$. In the current stage, we assume $\boldsymbol{A}$ is column normalized and define $\boldsymbol{B} = \boldsymbol{A}\boldsymbol{Q}_{\epsilon,n}$. The inner product between any two distinct columns $\boldsymbol{b}_i$ and $\boldsymbol{b}_j$ of $\boldsymbol{B}$, $\boldsymbol{b}_i^T\boldsymbol{b}_j$, normalized by their norms, $\boldsymbol{b}_i^T\boldsymbol{b}_i$ and $\boldsymbol{b}_j^T\boldsymbol{b}_j$, serves as the corresponding entry of the Gram matrix of $\boldsymbol{B}$. These three terms of interest are all composed of sums involving products of independent Gaussian variables and corresponding inner product in $\boldsymbol{A}$.

**Lemma 3.6.** *The product of two independent off-diagonal entries in the DogRot is zero-mean and follows a $\left(\sqrt{2}\epsilon, \sqrt{2}\epsilon\right)$ sub-exponential distribution* [2].

The proof is provided in Appendix C. Lemma 3.6 implies that the aforementioned terms are all sums of sub-exponential variables, and hence are themselves sub-exponential. Building upon this, the following theorem establishes a probabilistic bound for the inter-column coherence of $\boldsymbol{B}$.

**Theorem 3.7.** *Let $\boldsymbol{b}_i$ and $\boldsymbol{b}_j$ be two distinct columns in the effective sensing matrix mixed by the DogRot. Then, for all $\varepsilon \in (0,1)$, the coherence satisfies the following concentration inequality*

$$P\left\{\frac{|\boldsymbol{b}_i^T\boldsymbol{b}_j|}{\|\boldsymbol{b}_i\|_2\|\boldsymbol{b}_j\|_2} - \frac{(1-\epsilon)a_{ij}}{\epsilon(n-1)+1} \geq \varepsilon\right\} \leq 2\exp\left(-\frac{(\epsilon(n-1)+1)^2\varepsilon^2}{4\|\boldsymbol{G}_A - \boldsymbol{I}_n\|_F^2\epsilon^2}\right). \tag{7}$$

Appendix D details the derivation of this statement. Since our aim is to reduce the MC of HIM, we focus on bounding the right tail of this inequality. Theorem 3.7 reveals two key insights. Firstly, the expected coherence of $\boldsymbol{B}$ is a scaled-down version of that in $\boldsymbol{A}$, as the factor $\frac{1-\epsilon}{\epsilon(n-1)+1}$ is strictly less than 1. On the other hand, the probability of coherence deviating significantly from its mean decays exponentially with $n^2$, while the expectation itself tends to zero as $n \to \infty$. Together, these results ensure that, with high probability, large off-diagonal entries are suppressed, leading to an overall reduction in MC, which is precisely the objective in Problem 3.

---

[2] Note that a sub-exponential variable can have multiple valid pairs of parameters. The pair $\left(\sqrt{2}\epsilon, \sqrt{2}\epsilon\right)$ derived here is sufficient for our purposes.

### 3.2.3 THE ESSENCE OF DOGROT: PRESERVING SPARSITY

Since $n \gg p$, we can assume w.l.o.g. that Corollary 3.5 holds for DogRot when considering HIM. Therefore, applying the DogRot as a mixer transforms the relationship in (2) into the following form

$$\boldsymbol{y} \approx \boldsymbol{A}\boldsymbol{Q}_{\epsilon,n}\boldsymbol{Q}_{\epsilon,n}^T\boldsymbol{x}/\left(\epsilon(n-1)+1\right) = \frac{1}{(\epsilon(n-1)+1)}\boldsymbol{B}\boldsymbol{z}, \tag{8}$$

where $\boldsymbol{z} = \boldsymbol{Q}_{\epsilon,n}^T\boldsymbol{x}$. By Proposition 3.2, $\boldsymbol{Q}_{\epsilon,n}^T$ is also a DogRot. Assuming that $x_i = 0$ if $i \notin \mathcal{S}$, the transformed signal $\boldsymbol{z}$ takes the form

$$z_j = z_j + \sum_{i=1, i \neq j}^n q_{i,j} x_i = \begin{cases} \sum_{i \in \mathcal{S}} x_i q_{i,j} & , j \notin \mathcal{S}, \\ \sum_{i \in \mathcal{S}, i \neq j} x_i q_{i,j} & , j \in \mathcal{S}. \end{cases} \tag{9}$$

As a result, each $z_j$ becomes a Gaussian random variable. Specifically, if $j \notin \mathcal{S}$, $z_j \sim \mathcal{N}\left(0, \|\boldsymbol{x}\|_2^2 \epsilon\right)$; otherwise, i.e., $j \in \mathcal{S}$, $z_j \sim \mathcal{N}\left(x_j, \left(\|\boldsymbol{x}\|_2^2 - x_j^2\right)\epsilon\right)$. From a probabilistic perspective, the likelihood that $z_j$ exactly equals $x_j$ is zero. Therefore, the transformed signal $\boldsymbol{z}$ is no longer strictly sparse.

However, if $\epsilon$ is chosen to be sufficiently small compared to the magnitude of the nonzero coefficients in $\boldsymbol{x}$, then each $z_j$ remains close to $x_j$ in value. To recover the original sparsity structure, we can apply a hard thresholding operator $H_t$ to $\boldsymbol{z}$, defined as:

$$H_t(\boldsymbol{z}) = \begin{cases} z_j, & z_j > t, \\ 0, & z_j \leq t, \end{cases} \quad \text{for } 1 \leq j \leq n. \tag{10}$$

With an appropriate threshold, $H_t$ can filter out the original invalid indices and hence preserve the support of $\boldsymbol{x}$. As a result, the DogRot transformation satisfies the sparsity-preserving constraint in (3b) in a relaxed sense, by ensuring $\mathrm{supp}\left(H_t\left(\boldsymbol{Q}^{-1}\boldsymbol{x}\right)\right) = \mathrm{supp}\left(\boldsymbol{x}\right)$.

It is noteworthy that although multiplying the DogRot breaks the original strong correlations while preserving the support set, this comes at the cost of amplitude distortion in $\boldsymbol{z}$ relative to $\boldsymbol{x}$. This distortion is a random variable following $\mathcal{N}\left(0, \left(\|\boldsymbol{x}\|_2^2 - x_j^2\right)\epsilon\right)$. Hence, a trade-off arises: a small $\epsilon$ ensures better amplitude fidelity but weakens the decorrelation effect, whereas a large $\epsilon$ enhances decorrelation but may compromise amplitude accuracy and even support identifiability. According to Theorem 3.7, the tail probability of inter-column coherence decays exponentially with $\epsilon^2(n-1)^2$. Therefore, if $\epsilon$ is chosen too small, the tail decay is slow, limiting decorrelation; if too large, amplitude fidelity is severely degraded.

In practice, a moderate value of $\epsilon$ must be selected to balance decorrelation and fidelity. If a small $\epsilon$ is chosen properly, an effective reduction in MC can be achieved, and the distortion remains tolerable. This tolerance is justified for two primary reasons. First, the deviation is generally minor relative to the magnitude of $\boldsymbol{x}$. Second, as discussed previously, in the context of ill-conditioned problems, accurate support recovery is often more critical than exact amplitude recovery. Thus, the modest distortion is a reasonable price for the significant gain in support set estimation.

## 3.3 PRACTICAL IMPLEMENTATION

In the last subsection, we introduce a class of matrix referred to as DogRot and discuss their properties in the context of being used as mixers. The preceding analysis was conducted under the assumption that the highly ill-conditioned sensing matrix $\boldsymbol{A}$ is column-normalized. We will demonstrate that the proposed results and conclusions remain valid even $\boldsymbol{A}$ is not column-normalized.

If $\boldsymbol{A}$ is non-uniform, it can be decomposed as $\boldsymbol{A} = \hat{\boldsymbol{A}}\boldsymbol{W}$, where $\hat{\boldsymbol{A}}$ is the column-normalized version of $\boldsymbol{A}$ and $\boldsymbol{W}$ is a diagonal matrix representing the column norms of $\boldsymbol{A}$, i.e., $\boldsymbol{W} = \mathrm{diag}(\|\boldsymbol{a}_1\|_2, \|\boldsymbol{a}_2\|_2, \cdots, \|\boldsymbol{a}_n\|_2)$. Under this formulation, the relation in (2) becomes

$$\boldsymbol{y} \approx \hat{\boldsymbol{A}}\boldsymbol{Q}_{\epsilon,n}\boldsymbol{Q}_{\epsilon,n}^T\boldsymbol{W}\boldsymbol{x}/\left(\epsilon(n-1)+1\right) = \hat{\boldsymbol{A}}\boldsymbol{Q}_{\epsilon,n}\boldsymbol{Q}_{\epsilon,n}^T\boldsymbol{x}', \tag{11}$$

where $\boldsymbol{x}' = \boldsymbol{W}\boldsymbol{x}/\left(\epsilon(n-1)+1\right)$. Clearly, $\boldsymbol{x}'$ differs from $\boldsymbol{x}$ only in the amplitudes at the support set $\mathcal{S}$, and the original signal $\boldsymbol{x}$ can be recovered from $\boldsymbol{x}'$ using $\boldsymbol{W}^{-1}$. Thus, for a non-uniform sensing matrix, we can first factor out the power matrix $\boldsymbol{W}$ from $\boldsymbol{A}$, then apply the previously discussed framework to the scaled signal $\boldsymbol{x}'$.

---

**Algorithm 1** Sparse signal recovery with HIM

---

**Input** Observations $y$, HIM $A$, DogRot variance $\epsilon$, sparsity level $n_s$ or hard threshold $t$.
**Output** Estimated sparse signal $x$.
 1: Generate a DogRot matrix $Q_{\epsilon,n}$.
 2: Normalize the columns of $A$ to obtain $\hat{A}$ and compute the scaling matrix $W$.
 3: Form the mixed sensing matrix $\hat{A}Q_{\epsilon,n}$.
 4: Apply a standard sparse recovery algorithm to $y$ and $AQ_{\epsilon,n}$ to obtain $z$.
 5: Recover $x$ by inverting the transformation and applying hard thresholding:

$$x = (\epsilon(n-1)+1)\,W^{-1}H_t(Q_{\epsilon,n}z)$$

.

---

In conclusion, the proposed DogRot serves as an effective solution for solving the problem in (3), regardless of whether the HIM is column-normalized. The overall procedure for sparse signal recovery with a general HIM is outlined in Algorithm 1. Since the objective of this work is to enhance sparse signal recovery by indirectly manipulating the HIM, the proposed approach can be readily combined with any recovery algorithm, including optimization-based, greedy, and Bayesian methods. Moreover, although conventional techniques for reducing MC are ineffective for HIM due to its ill-conditioned structure, they can still be applied after DogRot to potentially yield further improvements, as demonstrated in Appendix F.

In this procedure, apart from the variance $\epsilon$ of the DogRot entries, the threshold $t$ used in the hard-thresholding function is also a key tuning parameter. If $\epsilon$ is appropriately chosen and the sparsity $n_s$ of $x$ is known, the original signal can be exactly recovered by retaining the $n_s$ largest entries in magnitude and setting the rest to zero, making the specific choice of $t$ less critical. Nevertheless, to ensure robustness in a more rigorous sense, one may set $t \geq 2.58\,\|x\|_2\,\sqrt{\epsilon}$. This choice guarantees that, with probability exceeding $0.99$, the perturbations introduced by the DogRot remain below the threshold and can be effectively filtered out. The constant $2.58$ arises from the properties of the Gaussian distribution, corresponding to the 99th percentile of the standard Gaussian.

Another advantage of the proposed DogRot is its minimal additional computational cost. Specifically, enhancing the original problem's properties via DogRot entails only three main operations: generating $\frac{n(n-1)}{2}$ Gaussian random numbers, performing a matrix multiplication with $A$ with complexity $\mathcal{O}(pn^2)$, and a signal detransformation step with complexity $\mathcal{O}(n^2)$. These operations are lightweight relative to the cost of solving the underlying sparse recovery problem and thus render DogRot a computationally efficient augmentation.

## 4 EXPERIMENTS

The problem of sparse signal recovery, particularly involving the HIM arises in a wide range of applications in signal processing and machine learning. To demonstrate the effectiveness of the proposed DogRot, we present test results in three example applications: MRI reconstruction, LSCM, and GDC in this paper. Due to space limitations, only the MRI reconstruction results are shown in this part. Relevant supplementary materials for MRI reconstruction and results of the other two applications are provided in the Appendix E, F, and G, respectively.

MRI reconstruction aims to recover the image of interest from under-sampled k-space data by exploiting the sparsity of the image in the wavelet or discrete cosine transform (DCT) domain (Lustig et al., 2007b; Eksioglu, 2016; Ryu et al., 2019; Ding et al., 2023). Although the discrete Fourier transform and DCT form orthogonal bases, the use of under-sampling masks often introduces aliasing into the sensing matrix, particularly with commonly used patterns such as Cartesian sampling. Furthermore, due to the inherently slow acquisition process, the sampling rate is typically low. Together, these factors result in an acquisition system that operates with a HIM.

### 4.1 EXPERIMENTAL SETTINGS

**Dataset:** In this paper, the test images are drawn from the publicly dataset available at `https://www.kaggle.com/datasets/masoudnickparvar/brain-tumor-mri-dataset` and the Shepp–Logan phantom. These images are resized to $128 \times 128$.

Table 1: Reduction in MC achieved through the proposed method in MRI reconstruction with different sampling patterns and rates

| Sampling rate (Cartesian) | 10% | 20% | 30% | 40% | 50% |
|---|---|---|---|---|---|
| $A$ | 0.99 | 0.90 | 0.74 | 0.99 | 0.99 |
| $AQ$ | 0.37 | 0.33 | 0.26 | 0.34 | 0.33 |
| **Sampling rate (Radial)** | **10%** | **20%** | **30%** | **40%** | **50%** |
| $A$ | 0.53 | 0.42 | 0.22 | 0.19 | 0.15 |
| $AQ$ | 0.29 | 0.21 | 0.19 | 0.13 | 0.11 |

Table 2: Comparison of PSNR before and after mixed by the proposed DogRot with 30% sampling rate. Bold values indicate improvements achieved by the proposed method.

| Sampling Rate: 30% (Cartesian) | Phantom | | Notumor | | Pituitary | |
|---|---|---|---|---|---|---|
| | $A$ | $AQ$ | $A$ | $AQ$ | $A$ | $AQ$ |
| **ISTA** | 16.26 | **18.19** | 12.31 | **14.62** | 12.17 | **14.79** |
| **ADMM** | 16.26 | **18.19** | 12.50 | **14.77** | 13.10 | **14.79** |
| **Sampling Rate: 30% (Cartesian)** | **Glioma** | | **Meningioma1** | | **Meningioma2** | |
| | $A$ | $AQ$ | $A$ | $AQ$ | $A$ | $AQ$ |
| **ISTA** | 15.29 | **17.16** | 13.12 | **15.05** | 15.06 | **16.98** |
| **ADMM** | 15.29 | **17.17** | 13.21 | **15.14** | 15.08 | **17.05** |
| **Sampling Rate: 30% (Radial)** | **Phantom** | | **Notumor** | | **Pituitary** | |
| | $A$ | $AQ$ | $A$ | $AQ$ | $A$ | $AQ$ |
| **ISTA** | 21.40 | **22.39** | 22.08 | **24.12** | 24.51 | **25.19** |
| **ADMM** | 21.40 | **22.39** | 22.15 | **24.12** | 24.60 | **25.22** |
| **Sampling Rate: 30% (Radial)** | **Glioma** | | **Meningioma1** | | **Meningioma2** | |
| | $A$ | $AQ$ | $A$ | $AQ$ | $A$ | $AQ$ |
| **ISTA** | 26.26 | **27.22** | 25.17 | **26.09** | 28.32 | **29.39** |
| **ADMM** | 26.26 | **27.25** | 25.21 | **26.12** | 28.92 | **30.42** |

**Sampling:** We consider the Cartesian and radial sampling patterns, which are commonly adopted in practical applications. The sampling rate varies between 10% and 50% in experiments to comprehensively test the performance of the proposed method. The sparsity in DCT domain is utilized in this paper.

**Metric:** We adopt the MC $\mu(A) \in [0, 1]$ to indicate the illness of the sensing matrix $A$ and the peak signal-to-noise (PSNR) presented in dB form to measure the reconstruction quality of the image of interest.

**Methods:** Two classical algorithms, ISTA and ADMM, are employed for MRI reconstruction. Using their performance without mixing as the baseline, we then evaluate the benefits of incorporating the proposed DogRot mixing. $\sqrt{\epsilon}$ is set to 0.025 in this experiment.

## 4.2 REDUCTION IN MC

Table 1 summarizes the MC of the sensing matrix before and after mixing under different sampling rate. In Table 1, $A$ denotes the MC of the original sensing matrix, while $AQ$ represents the MC after applying the DogRot mixing. As shown, the MC under Cartesian sampling typically reaches values as high as 0.99. By contrast, the proposed method reduces it to a substantially lower level. Moreover,

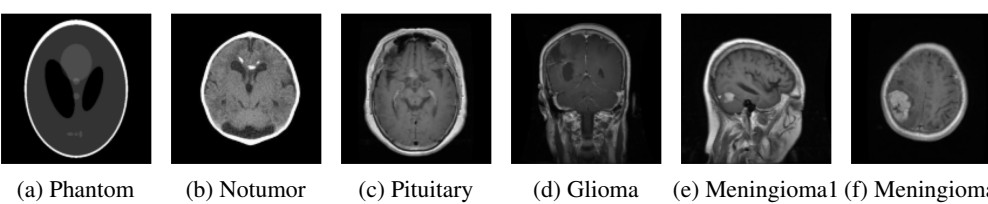

(a) Phantom     (b) Notumor     (c) Pituitary     (d) Glioma     (e) Meningioma1 (f) Meningioma2

Figure 1: Six different experimental MRIs acquired from the public dataset

even in scenarios where the initial MC is moderate, DogRot still achieves a noticeable reduction. In addition, the standard variation of the resulting MC remain on the order of 0.01, indicating the robustness and stability of the proposed approach.

### 4.3 IMPROVEMENT IN SIGNAL RECONSTRUCTION

We next demonstrate that the reduction in MC leads to improved sparse signal recovery. Taking the 30% sampling rate case for example, Table 2 provides the PSNR for six test images. These used images are illustrated in Figure 1. As presented, applying the proposed DogRot decorrelation leads to at least a 1 dB improvement in PSNR for both ISTA and ADMM in most cases. The gain is particularly pronounced under Cartesian sampling, where aliasing artifacts are severe. The results under 20% sampling rate and an illustrative example of the proposed method's effectiveness in mitigating aliasing are further provided in the Appendix E.

## 5 CONCLUSION

In this paper, we proposed a novel approach and formulated a new optimization problem aimed at reducing the MC of HIM matrices while preserving the solution to the original sparse signal recovery task. In particular, we introduced a class of DogRot matrix, as an effective and feasible solution of the proposed problem. The proposed DogRot exhibits several desirable properties: it reduces inter-column correlations when applied as a right-multiplier while approximately keeping the solution set of the original problem. Both theoretical analysis and experimental results confirm the effectiveness of the approach. Furthermore, the experiments also demonstrate significant improvements in sparse signal recovery performance.

Several directions merit further investigation. Although the proposed method enhances existing compressed sensing algorithms, these algorithms are not explicitly designed to exploit the transformed signal structure. Given that the signal entries become approximately Gaussian after applying the DogRot, developing customized Bayesian recovery algorithms could lead to additional performance gains. Moreover, the structure of DogRot could be generalized by drawing its entries from other sub-Gaussian or bounded distributions, potentially offering better trade-offs between coherence reduction and signal fidelity.

## ETHICS STATEMENT

This work focuses on developing a novel method, DogRot, to enhance sparse signal recovery in high-dimensional, under-determined systems. The primary applications considered, such as MRI reconstruction and other signal processing tasks, are technical and scientific in nature. All datasets used in this study are publicly available and were obtained in accordance with their respective usage policies. No new human or animal data were collected, and no personally identifiable information was used. While improved signal recovery techniques could theoretically be applied in sensitive contexts, our work is primarily methodological, focusing on algorithmic development. We encourage any application of DogRot in clinical or sensitive settings to comply with relevant ethical guidelines and data privacy regulations.

## REPRODUCIBILITY STATEMENT

All datasets used in this work, including test images for MRI reconstruction, sensing matrix generation for LSCM, and leukemia diagnosis data, are publicly available. The benchmark algorithms are well established in the literature and can be readily reproduced. Detailed descriptions of dataset sources, preprocessing steps, model parameters, and evaluation metrics are provided in the paper and Appendix. The DogRot implementation consists of a few simple steps and can be reproduced using standard Python libraries (NumPy, SciPy) or MATLAB. All experiments can be executed on a standard CPU-equipped machine with modest runtime requirements.

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

APPENDIX

## A PROOF OF THEOREM 3.3

*Proof.* **Theorem 3.3** can be proved using the Cramér-Chernoff method or by leveraging the properties of the scaled $\chi^2$ distribution (Vershynin, 2018; Tao, 2012; Van Handel, 2014). Specifically, following the Cramér-Chernoff method, we have

$$P\left\{\frac{\|\boldsymbol{q}_i\|_2^2}{\epsilon(n-1)+1} - 1 \geq \varepsilon\right\} \leq \min_{\lambda > 0} e^{-\lambda\varepsilon} \mathbb{E}\left[\exp\left(\lambda\left(\frac{\|\boldsymbol{q}_i\|_2^2}{\epsilon(n-1)+1} - 1\right)\right)\right]. \tag{12}$$

Representing $\epsilon(n-1)+1$ by $\alpha$, the right side of inequality (12) can be further arranged as

$$e^{-\lambda\varepsilon} \mathbb{E}\left[\exp\left(\lambda\left(\frac{\|\boldsymbol{q}_i\|_2^2}{\alpha} - 1\right)\right)\right] = e^{-\lambda\varepsilon}\mathbb{E}\left[\exp\left(\lambda\left(\frac{\sum_{j=1,j\neq i}^{n} q_{i,j}^2}{\alpha} - \frac{\alpha-1}{\alpha}\right)\right)\right]$$

$$= e^{-\lambda\left(\varepsilon + \frac{\alpha-1}{\alpha}\right)}\mathbb{E}\left[\prod_{j=1,j\neq i}^{n} \exp\left(\lambda\frac{q_{i,j}^2}{\alpha}\right)\right] = e^{-\lambda\left(\varepsilon + \frac{\alpha-1}{\alpha}\right)}\prod_{j=1,j\neq i}^{n}\mathbb{E}\left[\exp\left(\lambda\frac{q_{i,j}^2}{\alpha}\right)\right]. \tag{13}$$

Since $q_{i,j}$ follows $\mathcal{N}(0,\epsilon)$, the factor in (13) is computed as

$$\mathbb{E}\left[\exp\left(\lambda\frac{q_{i,j}^2}{\alpha}\right)\right] = \int_{-\infty}^{\infty} \frac{1}{\sqrt{2\pi\epsilon}}\exp\left(-\frac{q_{i,j}^2}{2\epsilon}\right)\exp\left(\lambda\frac{q_{i,j}^2}{\alpha}\right)dq_{i,j}$$

$$= \frac{1}{\sqrt{2\pi\epsilon}}\int_{-\infty}^{\infty}\exp\left(-\frac{(\alpha-2\epsilon\lambda)q_{i,j}^2}{2\epsilon\alpha}\right)dq_{i,j} = \sqrt{\frac{\alpha}{\alpha-2\epsilon\lambda}}. \tag{14}$$

Substituting equation (13) and (14) into inequality (12) yields

$$P\left\{\frac{\|\boldsymbol{q}_i\|_2^2}{\epsilon(n-1)+1} - 1 \geq \varepsilon\right\} \leq \min_{\lambda > 0} e^{-\lambda\left(\varepsilon + \frac{\alpha-1}{\alpha}\right)}\left(\frac{\alpha}{\alpha-2\epsilon\lambda}\right)^{\frac{n-1}{2}}. \tag{15}$$

According to the first-order optimality condition, the optimum of the right side of (12) is obtained when $\lambda = \frac{\alpha\varepsilon}{2\epsilon\left(\varepsilon + \frac{\alpha-1}{\alpha}\right)}$. As a result, we have

$$P\left\{\frac{\|\boldsymbol{q}_i\|_2^2}{\epsilon(n-1)+1} - 1 \geq \varepsilon\right\} \leq e^{-\frac{\alpha\varepsilon}{2\epsilon}}\left(1 + \frac{\alpha\varepsilon}{\alpha-1}\right)^{\frac{n-1}{2}}$$

$$= \exp\left(\frac{n-1}{2}\ln\left(1 + \frac{\alpha\varepsilon}{\alpha-1}\right) - \frac{\alpha\varepsilon}{2\epsilon}\right) = \exp\left(\frac{n-1}{2}\left(\ln\left(1 + \frac{\alpha\varepsilon}{\alpha-1}\right) - \frac{\alpha\varepsilon}{\alpha-1}\right)\right)$$

$$\leq \exp\left(\frac{n-1}{8}\left(\frac{\alpha}{\alpha-1}\right)^2\varepsilon^2\right) \leq \exp\left(\frac{(n-1)\varepsilon^2}{8}\right). \tag{16}$$

In a similar way, we can also obtain

$$P\left\{1 - \frac{\|\boldsymbol{q}_i\|_2^2}{\epsilon(n-1)+1} \geq \varepsilon\right\} \leq \exp\left(\frac{(n-1)\varepsilon^2}{8}\right). \tag{17}$$

Combining this two parts, we have

$$P\left\{\left|\frac{\|\boldsymbol{q}_i\|_2^2}{\epsilon(n-1)+1} - 1\right| \geq \varepsilon\right\} \leq 2\exp\left(-\frac{(n-1)\varepsilon^2}{8}\right), \tag{18}$$

which concludes the proof. $\qquad\square$

## B    PROOF OF THEOREM 3.4

*Proof.* Similar to the proof of **Theorem 3.3**, we can derive the following inequalities

$$P\left\{\frac{\|\boldsymbol{q}_i + \boldsymbol{q}_j\|_2^2}{2\epsilon(n-1)+2} - 1 \geq \varepsilon\right\} \leq \exp\left(-\frac{n\varepsilon^2}{8}\right), \tag{19}$$

$$P\left\{1 - \frac{\|\boldsymbol{q}_i - \boldsymbol{q}_j\|_2^2}{2\epsilon(n-1)+2} \geq \varepsilon\right\} \leq \exp\left(-\frac{n\varepsilon^2}{8}\right), \tag{20}$$

for all $\varepsilon \in (0,1)$. Combining the two inequalities, we have

$$P\left\{\frac{\boldsymbol{q}_i^T \boldsymbol{q}_j}{\epsilon(n-1)+1} \geq \varepsilon\right\} = P\left\{\frac{\|\boldsymbol{q}_i + \boldsymbol{q}_j\|_2^2}{2\epsilon(n-1)+2} - \frac{\|\boldsymbol{q}_i - \boldsymbol{q}_j\|_2^2}{2\epsilon(n-1)+2} \geq \varepsilon\right\}$$

$$\leq P\left\{\frac{\|\boldsymbol{q}_i + \boldsymbol{q}_j\|_2^2}{2\epsilon(n-1)+2} - 1 \geq \varepsilon\right\} + P\left\{1 - \frac{\|\boldsymbol{q}_i - \boldsymbol{q}_j\|_2^2}{2\epsilon(n-1)+2} \geq \varepsilon\right\}$$

$$= 2\exp\left(-\frac{n\varepsilon^2}{8}\right). \tag{21}$$

The negative part can be accordingly obtained and we eventually arrive at

$$P\left\{\left|\frac{\boldsymbol{q}_i^T \boldsymbol{q}_j}{\epsilon(n-1)+1}\right| \geq \varepsilon\right\} \leq 4\exp\left(-\frac{n\varepsilon^2}{8}\right). \tag{22}$$

□

## C    PROOF OF LEMMA 3.6

*Proof.* According to the result in (Simon, 2002), the product of two independent zero-mean Gaussian random variables with variances $\sigma_1^2$ and $\sigma_2^2$ follows the probability density function (PDF) and characteristic function (CF) given by

$$f_{qq}(x) = \frac{1}{\pi\sigma_1\sigma_2} K_0\left(\frac{|x|}{\sigma_1\sigma_2}\right), \Psi_{qq}(\omega) = \left(\frac{1}{1+\sigma_1^2\sigma_2^2\omega^2}\right)^{1/2} \tag{23}$$

where $K_v(z) = \int_0^\infty e^{-z\cosh t}\cosh(vt)\,dt$ is the modified Bessel function of the second kind (Olver, 2010; Idris et al., 2016).
This Lemma can be proved from both the moment generating function (MCF) and tail property perspective. We take the MCF for example. Based on the definition in (Vershynin, 2018), a variable x is $(v, \alpha)$ sub-exponential if

$$E\left[e^{\lambda(x-E[x])}\right] \leq \exp\left(\frac{v^2\lambda^2}{2}\right), \quad for \ |\lambda| < \frac{1}{\alpha}. \tag{24}$$

We can know from the symmetry that the expectation of the product of zero-mean Gaussian variables is zero. Moreover, from the CF in (23) and the definition of DogRot, we obtain the MCF for the product of two independent off-diagonal entries in the DogRot

$$M_{qq}(\lambda) = \frac{1}{\sqrt{1-\epsilon^2\lambda^2}} \leq \exp\left(\frac{v^2\lambda^2}{2}\right), \quad for \ |\lambda| < \frac{1}{\epsilon}. \tag{25}$$

We need to estimate an upper bound for this MCF. Taking the logarithm on both sides of the inequality yields $-\ln\left(1-\epsilon^2\lambda^2\right) \leq v^2\lambda^2$. Using the inequality $-\ln\left(1-x\right) \leq \frac{x}{1-x}$ for $0 < x < 1$, we obtain $-\ln\left(1-\epsilon^2\lambda^2\right) \leq \frac{\epsilon^2\lambda^2}{1-\epsilon^2\lambda^2}$. Consequently, we should find $v^2 \geq \frac{\sigma_1^2\sigma_2^2}{1-\sigma_1^2\sigma_2^2\lambda^2}$. If we set $\lambda^2 \leq \frac{1}{2\epsilon^2}$, i.e., $\alpha = \sqrt{2}\epsilon$, we can choose $v^2 = 2\epsilon^2$. Then, the product of two independent off-diagonal entries in the DogRot is shown to be $\left(\sqrt{2}\epsilon, \sqrt{2}\epsilon\right)$ sub-exponential.    □

## D  PROOF OF THEOREM 3.7

*Proof.* Unfolding $\boldsymbol{b}_i^T \boldsymbol{b}_j$ yields

$$\boldsymbol{b}_i^T \boldsymbol{b}_j = \left( \sum_{k=1}^n q_{k,i} \boldsymbol{a}_k \right)^T \left( \sum_{l=1}^n q_{l,j} \boldsymbol{a}_l \right) = \sum_{k=1}^n q_{k,i} q_{k,j} \boldsymbol{a}_k^T \boldsymbol{a}_k + \sum_{k=1}^n \sum_{l=1,l\neq k}^n q_{k,i} q_{l,j} \boldsymbol{a}_k^T \boldsymbol{a}_l, \quad (26)$$

Since $\boldsymbol{A}$ is assume to be column-normalized, each $\boldsymbol{a}_k$ satisfies $\boldsymbol{a}_k^T \boldsymbol{a}_k = 1$. According to Theorem 3.4, the first term concentrates around zero due to the asymptotic orthogonality of different columns in $\boldsymbol{Q}_{\epsilon,n}$. In the second term, $\boldsymbol{a}_k^T \boldsymbol{a}_l$ is fixed for a given $\boldsymbol{A}$ and we shortly denote $\boldsymbol{a}_k^T \boldsymbol{a}_l$ as $a_{kl}$ thereafter. Based on the definition in (4), we can expand the expression into

$$\boldsymbol{b}_i^T \boldsymbol{b}_j \approx \sum_{\substack{k=1,k\neq i}}^n \sum_{\substack{l=1,l\neq k,j \\ if \ k=j,l\neq i}}^n q_{k,i} q_{l,j} a_{kl} + \sum_{k=1,k\neq i,j}^n q_{k,i} a_{kj} + \sum_{l=1,l\neq i,j}^n q_{l,j} a_{il} + \left(1 - q_{j,i}^2\right) a_{ij}. \quad (27)$$

The first term in (27) is thus a sum of multiple independent, scaled $\left(\sqrt{2}\epsilon, \sqrt{2}\epsilon\right)$ sub-exponential random variables. By definition (Vershynin, 2018; Van Handel, 2014), sub-exponential variables exhibit heavier tails than sub-Gaussian variables. Given this, we can conservatively treat the second and third terms in equation (27) as a sum of independent, scaled, zero-mean $\left(\sqrt{2}\epsilon, 0\right)$ sub-exponential variables. In the fourth term, $q_{j,i}^2$ follows a scaled chi-squared distribution with one degree of freedom, which is a $(2\epsilon, 2\epsilon)$ sub-exponential variable with expectation $\epsilon$. Together, these yield that $\boldsymbol{b}_i^T \boldsymbol{b}_j$ is a sub-exponential variable concentrating around $(1 - \epsilon) a_{ij}$.

Similarly, the $\ell_2$ norm of an arbitrary column in $\boldsymbol{B}$ is expressed as follows

$$\boldsymbol{b}_i^T \boldsymbol{b}_i = \|\boldsymbol{q}_i\|_2^2 + 2 \sum_{k=1,k\neq i}^n \sum_{l=1,l<k,l\neq i}^n q_{k,i} q_{l,i} a_{kl} + 2 \sum_{k=1,k\neq i}^n q_{k,i} a_{ik}$$

$$\approx \epsilon(n-1) + 1 + 2 \sum_{k=1,k\neq i}^n \sum_{l=1,l<k,l\neq i}^n q_{k,i} q_{l,i} a_{kl} + 2 \sum_{k=1,k\neq i}^n q_{k,i} a_{ik}, \quad (28)$$

which is also a sub-exponential variable, with mean approximately $\epsilon(n-1) + 1$. As a result, the inter-column coherence of $\boldsymbol{B}$, given by $\frac{|\boldsymbol{b}_i^T \boldsymbol{b}_j|}{\|\boldsymbol{b}_i\|_2 \|\boldsymbol{b}_j\|_2}$, is expressed as a fraction involving three sub-exponential random variables. Furthermore, as observed in equations (27) and (28), these random variables share common components, leading to statistical dependence. Due to this nonlinear coupling and dependence structure, deriving a closed-form distribution or tight bound for the coherence is intractable. We therefore rely on approximation techniques to analyze the behavior of the modified coherence.

As shown in equations (27) and (28), the inner products of $\boldsymbol{B}$ involve summations of numerous chi-squared and Gaussian product random variables. Due to the complex and dependent structure of these terms, it is analytically intractable to characterize their exact distributions. Therefore, we resort to approximations and probabilistic inequalities to derive a tail bound for the inter-column coherence.

Firstly, we characterize the sub-exponential distribution of the inner products in $\boldsymbol{B}$ in more detail. According to the properties of sub-exponential random variables (Vershynin, 2018; Van Handel, 2014), the sub-exponential parameters for the mutual inner product $\boldsymbol{b}_i^T \boldsymbol{b}_j (i \neq j)$ are given by

$$v_{ij} = \sqrt{2\epsilon^2 \left( \sum_{(k,l)\in\mathcal{K}} a_{kl}^2 \right) + 4\epsilon^2 a_{ij}^2} = \sqrt{2\epsilon^2 \sum_{k=1}^n \sum_{l=1,l\neq k}^n a_{kl}^2} = \sqrt{2} \|\boldsymbol{G}_A - \boldsymbol{I}_n\|_F \epsilon, \quad (29)$$

$$\alpha_{ij} = \max \left\{ \max_{(k,l)\in\mathcal{K}} \sqrt{2} |a_{kl}| \epsilon, 2 |a_{ij}| \epsilon \right\} = \max \left\{ \sqrt{2} \mu(A) \epsilon, 2 |a_{ij}| \epsilon \right\}, \quad (30)$$

where $\mathcal{K} = \{(k,l) | 1 \leq k \leq n, 1 \leq l \leq n, k \neq l\} \setminus \{(i,j), (j,i)\}$. By the definition of HIM, i.e., the inter-column coherence follows a heavy-tail distribution truncated at 1, there are many pairs of $|a_{kl}|$ with value close to 1. Therefore, regardless the value of $|a_{ij}|$, $\max_{(k,l)\in\mathcal{K}} \sqrt{2} |a_{kl}| \epsilon$ would

approximately equal to $\sqrt{2}\mu\left(A\right)\epsilon$. In this case, when the original $i$-th and $j$-th column are highly correlated, i.e., $|a_{ij}|$ fall in the range $(\frac{\sqrt{2}}{2}, 1]$, the parameter $\alpha_{ij}$ equals to $2|a_{ij}|\epsilon$. Otherwise, $\alpha_{ij} = \sqrt{2}\mu\left(A\right)\epsilon$. Similarly, the sub-exponential parameters for the self-inner product $\boldsymbol{b}_i^T\boldsymbol{b}_i$ are given by

$$v_{ii} = \sqrt{2\epsilon^2 \sum_{k=1}^{n}\sum_{l=1,l\neq k}^{n} a_{kl}^2} = \sqrt{2}\|\boldsymbol{G}_A - \boldsymbol{I}_n\|_F\epsilon, \tag{31}$$

$$\alpha_{ii} = \max_{1\leq k<l\leq n}\sqrt{2}\,|a_{kl}|\,\epsilon = \sqrt{2}\mu\left(A\right)\epsilon. \tag{32}$$

We observe that the inner products between the columns of $\boldsymbol{B}$ are governed by two key indicators of the ill-conditioning of $\boldsymbol{A}$, i.e., the global coherence $\|\boldsymbol{G}_A - \boldsymbol{I}_n\|_F$ and the MC $\mu\left(A\right)$. In particular, the self-inner product $\boldsymbol{b}_i^T\boldsymbol{b}_i$ is a $(\sqrt{2}\|\boldsymbol{G}_A - \boldsymbol{I}_n\|_F\epsilon, \sqrt{2}\mu\left(A\right)\epsilon)$ sub-exponential variable with expectation $\epsilon(n-1)+1$. On the other hand, $\boldsymbol{b}_i^T\boldsymbol{b}_j$ is a $(\sqrt{2}\|\boldsymbol{G}_A - \boldsymbol{I}_n\|_F\epsilon, \max\left\{\sqrt{2}\mu\left(A\right)\epsilon, 2\,|a_{ij}|\,\epsilon\right\})$ sub-exponential variable with expectation $(1-\epsilon)a_{ij}$. Importantly, these distributions are not symmetric about the origin due to their non-zero means. As a result, directly analyzing the absolute normalized inner product $\frac{|\boldsymbol{b}_i^T\boldsymbol{b}_j|}{\|\boldsymbol{b}_i\|_2\|\boldsymbol{b}_j\|_2}$ is challenging. Instead, we first consider the original version $\frac{\boldsymbol{b}_i^T\boldsymbol{b}_j}{\|\boldsymbol{b}_i\|_2\|\boldsymbol{b}_j\|_2}$ before extending the result to the absolute value.

$\frac{\boldsymbol{b}_i^T\boldsymbol{b}_j}{\|\boldsymbol{b}_i\|_2\|\boldsymbol{b}_j\|_2}$ is still challenging to explicitly analyse due to the nonlinearity of the fraction and the statistical dependence between the numerator and denominators. To obtain a reasonable approximation, we employ the Delta method (Casella & Berger, 2002; Vaart, 1998), a standard technique in statistics, to approximate a nonlinear function of random variables using a Taylor expansion. Specifically, we define a function $\phi : \mathbb{R}^3 \mapsto \mathbb{R}$ as $\phi(\mathrm{x}, \mathrm{y}, \mathrm{z}) = \frac{\mathrm{x}}{\sqrt{\mathrm{y}}\sqrt{\mathrm{z}}}$, which maps three random variables to one. We further denote the means of input random variables as $\boldsymbol{\mu} = [\mu_{\mathrm{x}}, \mu_{\mathrm{y}}, \mu_{\mathrm{z}}]$. Since $\phi$ has continuous first partial derivatives away from the x-axis, the first-order Delta method yields

$$\phi(\mathrm{x}, \mathrm{y}, \mathrm{z}) \approx \phi(\mu_{\mathrm{x}}, \mu_{\mathrm{y}}, \mu_{\mathrm{z}}) + \left.\frac{\partial\phi}{\partial\mathrm{x}}\right|_{\boldsymbol{\mu}}(\mathrm{x} - \mu_{\mathrm{x}}) + \left.\frac{\partial\phi}{\partial\mathrm{y}}\right|_{\boldsymbol{\mu}}(\mathrm{y} - \mu_{\mathrm{y}}) + \left.\frac{\partial\phi}{\partial\mathrm{z}}\right|_{\boldsymbol{\mu}}(\mathrm{z} - \mu_{\mathrm{z}}), \tag{33}$$

where the partial derivatives at $\boldsymbol{\mu}$ are given by:

$$\left.\frac{\partial\phi}{\partial\mathrm{x}}\right|_{\boldsymbol{\mu}} = \frac{1}{\sqrt{\mu_{\mathrm{y}}}\sqrt{\mu_{\mathrm{z}}}},\ \left.\frac{\partial\phi}{\partial\mathrm{y}}\right|_{\boldsymbol{\mu}} = -\frac{\mu_{\mathrm{x}}}{2\sqrt{\mu_{\mathrm{z}}}\sqrt{\mu_{\mathrm{y}}}\mu_{\mathrm{y}}},\ \left.\frac{\partial\phi}{\partial\mathrm{z}}\right|_{\boldsymbol{\mu}} = -\frac{\mu_{\mathrm{x}}}{2\sqrt{\mu_{\mathrm{y}}}\sqrt{\mu_{\mathrm{z}}}\mu_{\mathrm{z}}}.$$

Since the involved quantities are sub-exponential variables concentrated around their means, a first-order approximation via the Delta method is adequate for our analysis. Taking the expectation on the both sides of equation (33), we have $\mathbb{E}[\phi(\mathrm{x}, \mathrm{y}, \mathrm{z})] \approx \phi(\mu_{\mathrm{x}}, \mu_{\mathrm{y}}, \mu_{\mathrm{z}})$. Applying this to our context, we can take $\boldsymbol{b}_i^T\boldsymbol{b}_j$, $\boldsymbol{b}_i^T\boldsymbol{b}_i$, and $\boldsymbol{b}_j^T\boldsymbol{b}_j$ as the inputs to $\phi$ and from earlier derivations, we have $\mu_{\mathrm{x}} = (1-\epsilon)a_{ij}$ and $\mu_{\mathrm{y}} = \mu_{\mathrm{z}} = \epsilon(n-1)+1$. Substituting into (33), we obtain:

$$\frac{\boldsymbol{b}_i^T\boldsymbol{b}_j}{\|\boldsymbol{b}_i\|_2\|\boldsymbol{b}_j\|_2} \approx \mu_{\phi} + \frac{1}{\mu_{\mathrm{y}}}\boldsymbol{b}_i^T\boldsymbol{b}_j - \frac{\mu_{\mathrm{x}}}{2\mu_{\mathrm{y}}^2}\boldsymbol{b}_i^T\boldsymbol{b}_i - \frac{\mu_{\mathrm{x}}}{2\mu_{\mathrm{y}}^2}\boldsymbol{b}_j^T\boldsymbol{b}_j, \tag{34}$$

where $\mu_{\phi} = \mathbb{E}[\frac{\boldsymbol{b}_i^T\boldsymbol{b}_j}{\|\boldsymbol{b}_i\|_2\|\boldsymbol{b}_j\|_2}] \approx \frac{\mu_{\mathrm{x}}}{\sqrt{\mu_{\mathrm{y}}}\sqrt{\mu_{\mathrm{z}}}} = \frac{(1-\epsilon)a_{ij}}{\epsilon(n-1)+1} < 1$. Leveraging the property of the sub-exponential variable again, we approximate $\frac{\boldsymbol{b}_i^T\boldsymbol{b}_j}{\|\boldsymbol{b}_i\|_2\|\boldsymbol{b}_j\|_2}$ as a sub-exponential random variable and its parameters are approximately given by

$$v^{\star} \approx \frac{\sqrt{\left(\epsilon\left(n-1\right)+1\right)^2 + \left(1-\epsilon\right)^2 a_{ij}^2}}{\left(\epsilon\left(n-1\right)+1\right)^2}\sqrt{2}\|\boldsymbol{G}_A - \boldsymbol{I}_n\|_F\epsilon \approx \frac{\sqrt{2}\|\boldsymbol{G}_A - \boldsymbol{I}_n\|_F\epsilon}{\left(\epsilon\left(n-1\right)+1\right)}, \tag{35}$$

$$\alpha^{\star} \approx \frac{1}{\left(\epsilon\left(n-1\right)+1\right)}\max\left\{\sqrt{2}\mu\left(A\right)\epsilon, 2\,|a_{ij}|\,\epsilon\right\} = \frac{1}{\left(\epsilon\left(n-1\right)+1\right)}\alpha_{ij}, \tag{36}$$

Having established $\mu_{\phi}$ as the expectation, our term of interest, $P\left\{\frac{\boldsymbol{b}_i^T\boldsymbol{b}_j}{\|\boldsymbol{b}_i\|_2\|\boldsymbol{b}_j\|_2} - \mu_{\phi} \geq \varepsilon\right\}$, is an one-side tail bound for the sub-exponential random variable $\frac{\boldsymbol{b}_i^T\boldsymbol{b}_j}{\|\boldsymbol{b}_i\|_2\|\boldsymbol{b}_j\|_2}$. We can exploit the Bernstein-type inequality (Vershynin, 2018; Van Handel, 2014) to provide an upper bound for this tail property:

$$P\left\{\frac{\boldsymbol{b}_i^T\boldsymbol{b}_j}{\|\boldsymbol{b}_i\|_2\|\boldsymbol{b}_j\|_2} - \mu_{\phi} \geq \varepsilon\right\} \leq \exp\left(-\min\left(\frac{\varepsilon^2}{2v^{\star 2}}, \frac{\varepsilon}{2\alpha^{\star}}\right)\right) \tag{37}$$

Table 3: Comparison of PSNR before and after mixed by the proposed DogRot with 20% sampling rate. Bold values indicate improvements achieved by the proposed method.

| Sampling Rate: 20%  (Cartesian) | Phantom | | Notumor | | Pituitary | |
|---|---|---|---|---|---|---|
| | **A** | **AQ** | **A** | **AQ** | **A** | **AQ** |
| **ISTA** | 15.03 | **16.08** | 11.71 | **12.95** | 11.82 | **13.30** |
| **ADMM** | 15.03 | **16.14** | 11.81 | **13.06** | 12.56 | **13.50** |

| Sampling Rate: 20%  (Cartesian) | Glioma | | Meningioma1 | | Meningioma2 | |
|---|---|---|---|---|---|---|
| | **A** | **AQ** | **A** | **AQ** | **A** | **AQ** |
| **ISTA** | 14.91 | **16.04** | 12.51 | **13.99** | 14.34 | **15.98** |
| **ADMM** | 14.91 | **16.12** | 12.60 | **14.12** | 14.37 | **13.50** |

| Sampling Rate: 20%  (Radial) | Phantom | | Notumor | | Pituitary | |
|---|---|---|---|---|---|---|
| | **A** | **AQ** | **A** | **AQ** | **A** | **AQ** |
| **ISTA** | 19.71 | **20.79** | 20.53 | **22.15** | 22.77 | **23.60** |
| **ADMM** | 19.71 | **20.79** | 20.55 | **22.15** | 22.78 | **23.62** |

| Sampling Rate: 20%  (Radial) | Glioma | | Meningioma1 | | Meningioma2 | |
|---|---|---|---|---|---|---|
| | **A** | **AQ** | **A** | **AQ** | **A** | **AQ** |
| **ISTA** | 24.36 | **25.22** | 22.95 | **24.88** | 25.19 | **26.11** |
| **ADMM** | 24.39 | **25.27** | 22.95 | **24.94** | 26.44 | **27.22** |

Since $\frac{\boldsymbol{b}_i^T \boldsymbol{b}_j}{\|\boldsymbol{b}_i\|_2 \|\boldsymbol{b}_j\|_2}$ is restricted between $-1$ and $1$, in the range we are interested in, the upper bound will be dominated by the quadratic term. Thus, the bound simplifies to:

$$P\left\{ \frac{\boldsymbol{b}_i^T \boldsymbol{b}_j}{\|\boldsymbol{b}_i\|_2 \|\boldsymbol{b}_j\|_2} - \mu_\phi \geq \varepsilon \right\} \leq \exp\left( -\frac{(\epsilon(n-1)+1)^2 \varepsilon^2}{4\|\boldsymbol{G}_A - \boldsymbol{I}_n\|_F^2 \epsilon^2} \right). \tag{38}$$

By symmetry, we can eventually arrive the following inequality

$$P\left\{ \frac{|\boldsymbol{b}_i^T \boldsymbol{b}_j|}{\|\boldsymbol{b}_i\|_2 \|\boldsymbol{b}_j\|_2} - \frac{(1-\epsilon)a_{ij}}{\epsilon(n-1)+1} \geq \varepsilon \right\} \leq 2\exp\left( -\frac{(\epsilon(n-1)+1)^2 \varepsilon^2}{4\|\boldsymbol{G}_A - \boldsymbol{I}_n\|_F^2 \epsilon^2} \right). \tag{39}$$

$\square$

# E  SUPPLEMENTARY MATERIALS FOR MRI RECONSTRUCTION

The PSNR comparison for six test images under a 20% sampling rate is provided in Table 3. Consistent with the 30% sampling case, the proposed method enhances the reconstruction quality achieved by existing algorithms, even under challenging conditions. Notably, as seen by comparing Tables 2 and 3, the PSNR obtained at 20% sampling with the proposed method can in some cases match or even approach that at 30% sampling achieved by the conventional method. This improvement indicates the potential of the proposed method to accelerate MRI data acquisition, further highlighting its practical advantage. A visual comparison of MRI reconstructions before and after applying the proposed method is provided in Figure 2. In this example, ADMM is employed to reconstruct MRIs under a 30% sampling rate with both Cartesian and radial sampling patterns. The pituitary and glioma images are selected as representative cases. In Figure 2, the label "1" corresponds to the image reconstructed by ADMM alone, while "2" denotes the result after incorporating the proposed method.

Figure 2a and Figure 2c illustrate that severe aliasing arises under Cartesian sampling when the sampling rate is low. After applying the proposed method, this aliasing is largely mitigated, as

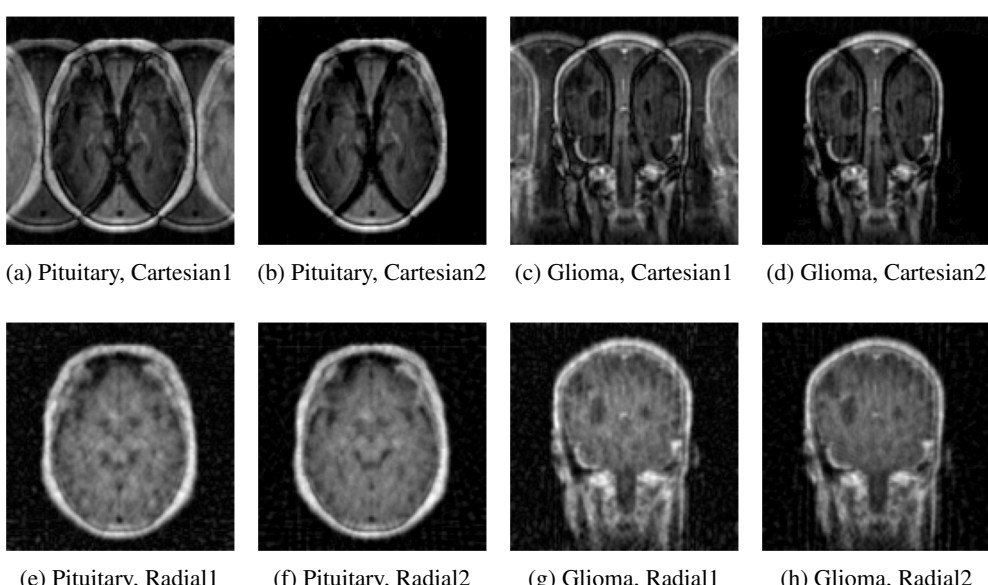

(a) Pituitary, Cartesian1  (b) Pituitary, Cartesian2  (c) Glioma, Cartesian1  (d) Glioma, Cartesian2

(e) Pituitary, Radial1  (f) Pituitary, Radial2  (g) Glioma, Radial1  (h) Glioma, Radial2

Figure 2: Comparison of images reconstructed by the ADMM and the proposed method under different sampling patter with 30% sampling rate

shown in Figure 2b and Figure 2d. In contrast, radial sampling inherently avoids strong aliasing, nevertheless, the proposed method is still able to enhance the reconstruction by recovering finer details in the MRI.

## F LOCALIZED STATISTICAL CHANNEL MODELING

Localized statistical channel modeling (LSCM) (Luo et al., 2023; Zhang et al., 2024; 2025) also serves as a concrete example of CS with HIM. In this scenario, the reference signal received power (RSRP), measured using a limited number of beams, is exploited to estimate the angular power spectrum (APS), which is always sparse in the spatial domain in high-frequency communication systems, at the base station (BS). Typically, fewer than 40 beams are available for RSRP measurements, while the APS is discretized over the spatial plane into $91 \times 72 = 6552$ grids. Moreover, due to the similarity among beams, their responses exhibit strong correlation. As a result, the corresponding $40 \times 6552$ sensing matrix is highly ill-conditioned.

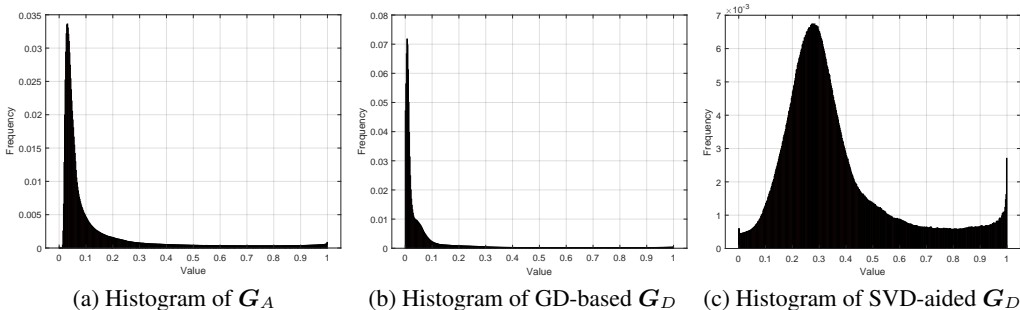

(a) Histogram of $\boldsymbol{G}_A$  (b) Histogram of GD-based $\boldsymbol{G}_D$  (c) Histogram of SVD-aided $\boldsymbol{G}_D$

Figure 3: Histograms of Gram matrices with before and after conventional projection

### F.1 Experimental Settings

**Dataset:** Zhang et al. (2024) provides a detailed procedure to generate diverse sensing matrices, which are used in the fifth-generation (5G) BS. In this part, we utilize $A$ generated by the method in (Zhang et al., 2024) to thoroughly evaluate the effectiveness of the proposed DogRot. We represent the BS status by a triple $(x, y, z)$, where $x$ and $y$ denote the tilt and azimuth, respectively, and $z$ specifies the beam pattern selected to satisfy coverage requirements. APS is randomly generate to examine the reconstruction performance. We include multiple BS parameter settings path number scenarios in this part.

**Metric:** We adopt the MC $\mu(A) \in [0, 1]$ to indicate the illness of the sensing matrix $A$. Two key metrics are selected to assess the recovery quality. The first one is the normalized mean square error (NMSE), which quantifies the deviation between the recovered APS and the ground truth. In addition, since the sparse signal $x$ is a flattened vector of size 6552 obtained from a $91 \times 72$ two-dimensional APS, we employ the Earth Mover's Distance (EMD), also known as the Wasserstein distance, to better capture the structural similarity between the recovered and true APS distributions. Formally, for a metric space $(\mathcal{X}, d)$ and probability measures $\mu$ and $\nu$ defined on $\mathcal{X}$, the $p$-Wasserstein distance is defined as

$$W_p(\mu, \nu) = \left( \inf_{\gamma \in \Pi(\mu, \nu)} \int_{\mathcal{X} \times \mathcal{X}} d(x, y)^p \, d\gamma(x, y) \right)^{1/p} \tag{40}$$

where $\Pi(\mu, \nu)$ denotes the set of all couplings of $\mu$ and $\nu$, i.e., the set of all joint distributions $\gamma$ on $\mathcal{X} \times \mathcal{X}$ with marginals $\mu$ and $\nu$. In our case, we adopt the Euclidean distance for $d(x, y)$. While NMSE serves as a classical error metric, EMD provides a more perceptually meaningful measure of discrepancy, reflecting the spatial distortion between the ground truth and the reconstructed APS.

**Methods:** In the content of channel estimation, coarse prior knowledge about the number of paths $K$, i.e., the sparsity of $x$, is typically available, and the APS is inherently nonnegative. Hence, the greedy non-negative OMP (NNOMP) algorithm and weighted NNOMP (WNNOMP) tailored by Zhang et al. (2024) are adopted for experiments. ADMM is also included as a widely used benchmark for this class of problems. $\sqrt{\epsilon}$ is set to 0.025 in this experiment.

### F.2 Reduction in mutual coherence

We first take a (2, 243, 0) $A$ for example to show the illness of the HIM and the failure encountered by existing projection methods. It is an extremely fat matrix with $\mu(A)$ reaching up to 0.99999986. The statistical histogram of the off-diagonal entries in its Gram matrix $G_A$ is shown in Figure 3a. As evident from the figure, the distribution exhibits a heavy tail as values approach 1, reflecting the strong correlations among columns.

The performance of two existing method to reduce $\mu(A)$ is depicted in Figure 3b and Figure 3c. It is revealed that the GD-based projector (Abolghasemi et al., 2010) can partially decrease the inter-correlation of $A$, however, the heavy-tail property of values in $G_D$ persists. This indicates that the expectation to reduce the MC through a global reduction is ineffective when applied to the HIM. Moreover, the property of projected sensing matrix is even worse with the SVD-based projector (Elad, 2007). This performance deterioration is likely attributable to the extreme ill-conditioning of $A$, which impairs the stability and effectiveness of low-rank approximations. Quantitatively, the MC is changed from 0.99999986 to 0.9999995 and 0.9999999998 after GD's and SVD's projection, respectively, which is marginal. In summary, existing techniques aimed at manipulating the sensing matrix by imposing a projector on the left fail to yield meaningful improvements in the presence of HIM. This underscores the necessity of the novel method proposed in this paper, which is specifically tailored to the HIM challenging scenario.

Table 4 summarizes the MC of the sensing matrices reduced by the proposed method with $\sqrt{\epsilon}$ set to 0.025. In particular, the label $A$ denotes the MC of the original sensing matrix while $AQ$ and $PAQ$ represent the results after applying the proposed DogRot mixing and further incorporating a conventional projector, respectively. As shown, unlike conventional direct projection approaches, DogRot can effectively decorrelate HIM even in extreme cases where the columns are nearly parallel. Furthermore, combining a projection with the proposed mixing achieves an additional reduction in MC. This demonstrates the versatility of DogRot, as it can be seamlessly integrated not only with signal recovery algorithms but also with existing coherence reduction techniques.

To further illustrate the decorrelation effect, histograms, CDFs, and heatmaps of the Gram matrices

Table 4: Reduction in MC achieved through the proposed mixing strategy.

| Parameter | (2, 243, 0) | (4, 240, 0) | (5, 240, 0) | (7, 127, 0) |
|---|---|---|---|---|
| $A$ | 0.99 | 0.99 | 0.99 | 0.99 |
| $AQ$ | 0.90 | 0.90 | 0.90 | 0.90 |
| $PAQ$ | 0.79 | 0.78 | 0.79 | 0.79 |
| **Parameter** | **(8, 103, 0)** | **(8, 314, 0)** | **(6, 141, 1)** | **(6, 214, 1)** |
| $A$ | 0.99 | 0.99 | 0.99 | 0.99 |
| $AQ$ | 0.90 | 0.90 | 0.90 | 0.90 |
| $PAQ$ | 0.79 | 0.79 | 0.80 | 0.80 |

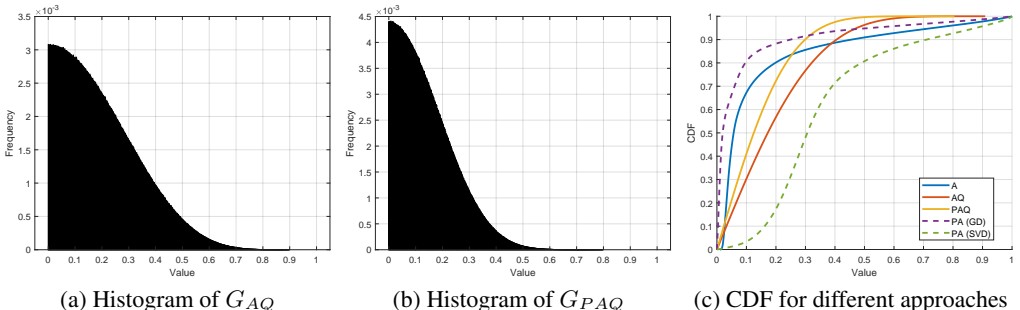

(a) Histogram of $G_{AQ}$      (b) Histogram of $G_{PAQ}$      (c) CDF for different approaches

Figure 4: Histograms and CDFs for the Gram matrices after mixing and projection

corresponding to $A$, $AQ$, and $PAQ$ are presented in Figure 4 and Figure 5, respectively. Compared with Figure 3a, Figure 4 demonstrates that the heavy tail in the Gram matrix is significantly suppressed, and the majority of column correlations in the sensing matrix are reduced to below 0.5. The CDFs and heatmaps further confirm this improvement by visually highlighting the reduction in inter-column coherence.

## F.3 Improvement in Signal Reconstruction

We have demonstrated the effectiveness of the proposed DogRot in reducing the MC of the effective sensing matrix, supported by both theoretical analysis and experimental validation. Then we show that this reduction in MC correspondingly enhances the estimation of the sparse APS. The APS estimation performance under various azimuth, tilt, and coverage scenario configurations is summarized in Table 5. Table 5 shows that the proposed DogRot-based mixing (and subsequent

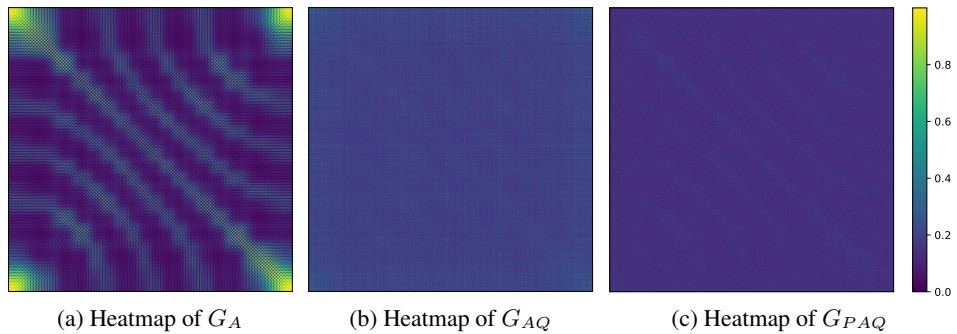

(a) Heatmap of $G_A$      (b) Heatmap of $G_{AQ}$      (c) Heatmap of $G_{PAQ}$

Figure 5: Heatmap for the Gram matrices before and after mixing and projection

Table 5: Comparison of APS estimation performance before and after mixing and projection. Bold values indicate improvements achieved by the proposed method. The best performance across different recovery methods for each setting and metric is underlined.

| $(2, 243, 0)$, $K = 5$ | NMSE | | | EMD | | |
|---|---|---|---|---|---|---|
| | $A$ | $AQ$ | $PAQ$ | $A$ | $AQ$ | $PAQ$ |
| **NNOMP** | 0.97 | **0.92** | **0.92** | 29.47 | **24.02** | **23.78** |
| **WNNOMP** | 42.75 | **0.97** | **0.96** | 23.81 | 24.22 | **23.46** |
| **ADMM** | 0.96 | **0.84** | **0.84** | 27.17 | **22.31** | **22.31** |

| $(2, 243, 0)$, $K = 10$ | NMSE | | | EMD | | |
|---|---|---|---|---|---|---|
| | $A$ | $AQ$ | $PAQ$ | $A$ | $AQ$ | $PAQ$ |
| **NNOMP** | 1.01 | **0.99** | **0.98** | 32.76 | **22.09** | **20.83** |
| **WNNOMP** | 113.52 | **1.04** | **1.03** | 27.27 | **22.59** | **22.28** |
| **ADMM** | 1.00 | **0.97** | **0.97** | 32.95 | **25.55** | **25.55** |

| $(4, 240, 0)$, $K = 5$ | NMSE | | | EMD | | |
|---|---|---|---|---|---|---|
| | $A$ | $AQ$ | $PAQ$ | $A$ | $AQ$ | $PAQ$ |
| **NNOMP** | 0.98 | **0.94** | **0.93** | 29.78 | **24.12** | **23.28** |
| **WNNOMP** | 48.15 | **0.95** | **0.95** | 23.15 | 24.43 | **23.08** |
| **ADMM** | 0.88 | **0.81** | **0.81** | 27.13 | **22.28** | **22.28** |

| $(4, 240, 0)$, $K = 10$ | NMSE | | | EMD | | |
|---|---|---|---|---|---|---|
| | $A$ | $AQ$ | $PAQ$ | $A$ | $AQ$ | $PAQ$ |
| **NNOMP** | 1.00 | **0.98** | **0.98** | 32.94 | **21.63** | **22.10** |
| **WNNOMP** | 119.49 | **1.01** | **1.01** | 27.46 | **21.61** | **21.55** |
| **ADMM** | 0.99 | **0.97** | **0.97** | 32.69 | **24.82** | **24.46** |

| $(6, 141, 1)$, $K = 5$ | NMSE | | | EMD | | |
|---|---|---|---|---|---|---|
| | $A$ | $AQ$ | $PAQ$ | $A$ | $AQ$ | $PAQ$ |
| **NNOMP** | 0.99 | **0.95** | **0.93** | 29.36 | **24.02** | **23.78** |
| **WNNOMP** | 141.48 | **0.99** | **0.99** | 27.14 | **25.92** | **25.53** |
| **ADMM** | 0.92 | **0.90** | **0.90** | 22.53 | **15.90** | **15.90** |

| $(6, 141, 1)$, $K = 10$ | NMSE | | | EMD | | |
|---|---|---|---|---|---|---|
| | $A$ | $AQ$ | $PAQ$ | $A$ | $AQ$ | $PAQ$ |
| **NNOMP** | 1.06 | **1.02** | **1.00** | 29.29 | **21.51** | **21.04** |
| **WNNOMP** | 157.47 | **1.05** | **1.03** | 31.71 | **20.66** | **20.34** |
| **ADMM** | 0.92 | **0.79** | **0.79** | 32.95 | **25.55** | **25.55** |

projection) substantially improves the reconstruction quality in most scenarios. To comprehensively evaluate performance, we consider three BS parameter settings and two path number scenarios. In terms of NMSE, the incorporation of the mixer and projector typically yields performance gains in the range of 1% - 12.5%. For the EMD, improvements of up to 36.4% is observed. While the NMSE values remain relatively high, the significant reduction in EMD suggests that the proposed approach enables existing recovery algorithms to produce a more accurate index estimation and hence an APS that more closely resemble the ground truth.

It is important to acknowledge the intrinsic difficulty of the HIM-based problem, particularly for

such an ill-conditioned settings with $\boldsymbol{A} \in \mathbb{R}^{40 \times 6552}$ in LSCM. Such configurations represent extremely ill-conditioned systems that are notoriously challenging to solve in practice. This is evident from the baseline results using the original sensing matrix $\boldsymbol{A}$, where the NMSE consistently hovers around 1, and the EMD remains high, both indicating ineffective recovery performance. Consequently, the performance of the proposed method represents a meaningful shift, from near-intractable recovery to a substantially more faithful solution. Notably, the improvement becomes more pronounced as the sparsity level increases, aligning with the insight from Equation **??** that lower MC enables recovery of signals with higher sparsity.

A weird phenomena here is that a better index estimation not directly leads to a better magnitude estimation. This stems from the significant norm fluctuation among the columns of $\boldsymbol{A}$ in LSCM, as analyzed in (Zhang et al., 2024). Therefore, the columns with similar direction may have largely different norms, which severely confuse the estimation of accurate index and the corresponding coefficient. However, if a better index estimation is acquire by the proposed method, many post-processing techniques can be applied to further calibrate the strength of a certain path, such as focusing the beam or power detection.

Additional insights can be drawn by examining the behavior of different recovery algorithms. Notably, the WNNOMP method proposed in (Zhang et al., 2024), although specifically tailored for the original HIM, yields a high NMSE, indicating poor magnitude estimation. However, its EMD is relatively lower than that of other methods, suggesting that WNNOMP is more effective at capturing the spatial structure of the APS despite inaccuracies in amplitude. This observation implies that WNNOMP, in its original form, lacks robustness in magnitude estimation when applied to ill-conditioned sensing matrices. Encouragingly, the application of the proposed DogRot-based mixing significantly stabilizes magnitude recovery, while further enhancing the overall reconstruction performance. As a result, WNNOMP combined with the proposed approach generally achieves competitive or superior performance compared to other baseline methods.

## G  GENE-BASED DISEASE CLASSIFICATION

Another representative example of CS with HIM is disease classification based on gene expression monitoring with DNA microarrays (Golub et al., 1999; Zou & Hastie, 2005; Li & Li, 2008). A typical microarray dataset contains thousands of genes but fewer than 100 samples, making the sensing matrix extremely ill-conditioned. Many existing works attempt to address this issue by feature elimination, relying on expert knowledge or statistical tools (Guyon et al., 2002) to reduce the number of genes used in the final model. However, such preprocessing may lead to the loss of critical features and information. Only a few approaches can directly tackle this inverse problem and select genes in a satisfactory manner. In this work, we further evaluate the performance of the proposed method in this particularly challenging application.

### G.1  EXPERIMENTAL SETTINGS

**Dataset:** We evaluate the proposed method using the well-known leukemia dataset, which contains 7129 genes and 72 samples (Golub et al., 1999). Another challenge is that is that many genes exhibit highly similar expression patterns, leading to strong correlations and consequently high MC in the expression matrix. This dataset, publicly available at `https://github.com/pulkitmehta/Molecular-Cancer-Classification-by-ML`, includes two class of cancers, i.e., acute myeloid leukemia (AML) and acute lymphoblastic leukemia (ALL). The objective is to construct a diagnostic rule based on the expression levels of the 7129 genes to predict the type of leukemia. While the dataset is commonly divided into a 38-sample training set and a 34-sample test set, we instead combine all samples and investigate the effectiveness of the proposed DogRot method by varying the number of training samples. This variation allows us to simulate a range of practical scenarios with different levels of data scarcity and challenge.

**Metric:** We adopt the MC $\mu(\boldsymbol{A}) \in [0, 1]$ to indicate the illness of the sensing matrix $\boldsymbol{A}$ and the classification accuracy to indirectly reflect the quality of gene selection.

**Methods:** Two classical regression models are adopted. The first one is $\ell_1$-regularized regression, i.e., LASSO, while the second is the elastic net (Zou & Hastie, 2005), which combines both $\ell_1$ and $\ell_2$ regularization. $\sqrt{\epsilon}$ is set to 0.025 in this experiment.

Table 6: Reduction in MC achieved through the proposed mixing with different training sample number

| Training Sample Number | 15 | 20 | 25 | 30 |
|:---:|:---:|:---:|:---:|:---:|
| $A$ | 0.99 | 0.99 | 0.99 | 0.99 |
| $AQ$ | 0.96 | 0.94 | 0.91 | 0.89 |

Table 7: Comparison of prediction accuracy before and after mixed by the proposed DogRot. Bold values indicate improvements achieved by the proposed method.

| Training Sample Number | 15 | | 20 | |
|:---:|:---:|:---:|:---:|:---:|
| | $A$ | $AQ$ | $A$ | $AQ$ |
| $\ell_1$ **regularization** | 53.33% | **72.04%** | 73.19% | **82.29%** |
| **elastic net** | 75.95% | **83.46%** | 82.33% | **87.71%** |

| Training Sample Number | 25 | | 30 | |
|:---:|:---:|:---:|:---:|:---:|
| | $A$ | $AQ$ | $A$ | $AQ$ |
| $\ell_1$ **regularization** | 82.19% | **86.73%** | 83.21% | **87.13%** |
| **elastic net** | 83.71% | **91.85%** | 86.72% | **94.62%** |

## G.2 REDUCTION IN MUTUAL COHERENCE

Table 6 records the reduction in MC of the expression matrix achieved by the proposed method. Notably, even in the challenging case of a $15 \times 7129$ $A$, the MC $\mu(A)$ can still be effectively reduced. As the number of training samples increases, the reduction becomes more pronounced.

## G.3 IMPROVEMENT IN CLASSIFICATION ACCURACY

Table 7 presents the classification accuracy obtained using two regression models. The results demonstrate that the proposed method substantially enhances classification accuracy compared with standard regression, with the improvement being especially remarkable when the number of training samples is small. Furthermore, the elastic net consistently outperforms LASSO, owing to its ability not only to enforce sparsity but also to perform simultaneous shrinkage and selection. This property enables the elastic net to generate grouped solutions, which better reflect the cooperative functional mechanisms underlying DNA expression.

