# OpenReview forum: "DogRot: Taming Highly Ill-Conditioned Sensing Matrix in Sparse Signal Recovery by Random Gaussian Rotator"
_ICLR.cc/2026/Conference — Submitted to ICLR 2026_

### Official Review · Reviewer_qyrv · 2025-10-31

**Soundness:** 3
**Presentation:** 2
**Contribution:** 2
**Rating:** 2
**Confidence:** 3

**Summary:**

The paper proposes a preconditioned for severely ill posed matrices in compressed sensing to improve recover performance. The preconditioned is randomized.

**Strengths:**

The numerical results are compelling in terms of the improvements on matrix metrics and performance.

**Weaknesses:**

The paper imputes changes to the signal $x$ to $z = Q^{-1} x$ but does not account for the mismatch in performance in recovery between the two. The latter is less sparse and may not yield the same recovery performance as the former, so in practice there is a tradeoff between the improvements in the CS matrix and the distortion of the recoverable signal.

The problem instantiated in (3) makes the preconditioning specific to the support of the signal $x$.

Although the paper mentions the notion of measuring either recovery of the signal or its support, there is no such evaluation in the experimental results.

The results on performance are focused on a single matrix aspect ratio.

The experimental results are somewhat underwhelming in their extent (six images). It is not clear why the large number of additional images from the same Kaggle dataset were not used; other resources like https://openneuro.org can provide larger datasets for validation.

**Questions:**

In line 91, what is $\hat{\mathbf A}$? Is it the column-normalized version of $\mathbf A$ that is defined in page 7? What is $G_A$? (The Gram matrix of $A$?)

Why can you safely assume in (8) that $Q^{-1} \approx c\cdot Q^T$? This does not necessarily hold in non-asymptotic regimes (Corollary 3.5).

In the right hand side of (9), should $z_j$ be $x_j$? Is an $x_j$ missing from the bracketed expression?

How does one obtain the  "appropriate threshold" of line 293?

Why is $\mathbf PA$ not included as a baseline in the comparison of Table 5?

Why is EMD a good distance to use in Section F.1?

---

> ### Author Response · Authors · 2025-11-24
> **Response to Reviewer qyrv (Part I)**
>
> ## Question 1
> Thank you for pointing out the unclear definitions of $\hat A$ and $G_A$. As understood, $\hat A$ and $G_A$ denote the column-normalized version and Gram matrix of $A$, respectly. The definition of $\hat A$ is acctually delivered before line 91 and page 7 at the footnote of line 48. However, we agree the placement is not prominent. We will move the definition into the main text and add the missing definition of $G_A$ in the revision.
>
> ## Question 2
> Many thanks for the reviewer's concern on the soundness of replacing $Q_{\epsilon,n}^{-1}$ by $\frac{1}{\epsilon \left( {n - 1} \right) + 1}{Q}_{\epsilon,n}^T$ in the non-asymptotic regimes. We have provided a detailed theoretical analysis on the deviation between these two terms and the corresponding effect on signal recovery. Since the derivation and discussion for these issues are extremely long and hard to display in a markdown environment. We summarize these analysis in [anonymous link](https://anonymous.4open.science/r/ICLR-2026-Rebuttal-AF33). Please refer to Figures from '01.png' to '05.png'.
>
> ## Question 3
> Thanks for the reviewer's question about the equation (9) in the manuscript. We have corrected this typo in the revised manuscript.
>
> ## Question 4
> Many thanks for the reviewer's concern on a practical rule for determining the hyper-parameters involved in this work. Since the derivation is dense, we also include this discussion in [anonymous link](https://anonymous.4open.science/r/ICLR-2026-Rebuttal-AF33) in Figure '06.png' to '07.png'.
>
> ## Question 6
> Thank you for the question regarding the use of EMD as a metric for evaluating APS estimation. In this work, we employ both NMSE and EMD to assess APS recovery performance. NMSE becomes small only when both the support of the APS and the corresponding coefficients are accurately recovered. However, NMSE fails to reflect partial improvements when the estimated APS is not perfectly accurate, e.g., when angles are approximately recovered but slightly shifted. To address this limitation, we introduce EMD as a complementary metric.
>
> EMD measures the minimum cost required to transport one distribution into another capturing not only the difference in probability mass between two distributions, but also the spatial displacement required to align them. As such, EMD takes into account both support accuracy and coefficient accuracy in a unified manner. In contrast, the top-k support precision suggested by Reviewer J23z evaluates only support recovery. It provides no meaningful distinction when support is coarsely estimated or when coefficients deviate from the true values. Moreover, once the support is exactly identified, top-k precision remains constant and is insensitive to further improvements.
>
> It is important to emphasize that the LSCM problem considered in this work is extremely ill-conditioned due to the limited available observations and the high angular resolution required. This leads to a severely ill-conditioned sensing matrix. As shown in Table 5, the problem is nearly unsolvable for conventional methods even with as few as five propagation paths. With the proposed approach, although the estimated angles may not perfectly match the true supports, they are often recovered within the correct vicinity, as illustrated by Figure 'APS' in [anonymous link](https://anonymous.4open.science/r/ICLR-2026-Rebuttal-AF33).
>
> This figure illustrates the APS recovery results in an experiment using NNOMP under the base station configuration (2, 243, 0), assuming 5 resolvable propagation paths. As shown, the APS recovered by the conventional method deviates significantly from the ground truth, whereas the angles estimated by the proposed method largely remain in the vicinity of their true orientations. This observation highlights that, without the proposed preconditioning, the conventional method struggles to obtain even a coarse reconstruction resembling the ground truth. Although the proposed method still cannot achieve fully accurate APS recovery in this highly challenging setting, it enables a meaningful transition from an almost unsolvable problem to an approximate but informative reconstruction.
>
> In the scenario shown, evaluating the NMSE of the two recovered APS signals yields similarly large values, offering little insight into the relative improvement. The support recovery accuracy also reflects only a modest gain of about 20\%. However, the adopted EMD metric effectively captures the improvement in the distributional similarity between the estimated and true APS. Therefore, EMD is the most appropriate metric for depicting the advantage provided by the proposed method in such highly ill-conditioned LSCM scenarios.

---

> ### Author Response · Authors · 2025-11-24
> **Response to Reviewer qyrv (Part II)**
>
> ## Question 5
> We appreciate the reviewer’s concern regarding the absence of the conventional left-multiplying preconditioning method "$PA$" as a baseline in Table 5 for angular power spectrum (APS) estimation. In the submitted manuscript, we did not include this method because we had already demonstrated that left-multiplying preconditioning fails to effectively reduce the mutual coherence of the highly ill-conditioned sensing matrix used in LSCM. As a result, any improvement in downstream recovery metrics—such as APS estimation accuracy—is inherently limited. Our experimental results further confirm this observation, i.e., applying the "$PA$" method yields negligible performance gains compared with the unprocessed sensing matrix. Based on this, we considered the method ineffective for the problem setting and thus omitted it as a benchmark for conciseness.
>
> For completeness, we provide the LSCM comparison results including the conventional left-multiplying $\mathbf{PA}$ method in the following tables. Consistent with our theoretical analysis, applying $\mathbf{PA}$ directly to the highly ill-conditioned sensing matrix yields only marginal improvements in sparse recovery. This limitation precisely motivates our development of a right-multiplying preconditioner in this work. Nevertheless, once the ill-conditioning is partially alleviated by the proposed method, the conventional left-multiplying $\mathbf{PA}$ technique can indeed offer additional performance gains.
>
> ##### (2, 243, 0), K=5
>
> | Method | A (NMSE) | **PA** (NMSE) | AQ (NMSE) | PAQ (NMSE) | A (EMD) | **PA (EMD)** | AQ (EMD) | PAQ (EMD) |
> |-------|----|---------|-----|------|----------|--------------|------------|-------------|
> | **NNOMP** | 0.97 | 0.96 | **0.92** | **0.92** | 29.47 | 27.63 | **24.02** | **23.78** |
> | **WNNOMP** | 42.75 | 23.15 | **0.97** | **0.96** | 23.81 | 23.69 | 24.22 | **23.46** |
> | **ADMM** | 0.96 | 0.90 | **0.84** | __**0.84**__ | 27.17 | 24.23 | **22.31** | __**22.31**__ |
>
> ##### (2, 243, 0), K=10
>
> | Method | A (NMSE) | **PA** (NMSE) | AQ (NMSE) | PAQ (NMSE) | A (EMD) | **PA (EMD)** | AQ (EMD) | PAQ (EMD) |
> |-------|----|---------|-----|------|----------|--------------|------------|-------------|
> | **NNOMP** | 1.01 | 1.01 | **0.99** | **0.98** | 32.76 | 30.18 | **22.09** | __**20.83**__ |
> | **WNNOMP** | 113.52 | 30.24 | **1.04** | **1.03** | 27.27 | 26.15 | **22.59** | **22.28** |
> | **ADMM** | 1.00 | 1.00 | **0.97** | __**0.97**__ | 32.95 | 30.06 | **25.55** | **25.55** |
>
> ##### (4, 240, 0), K=5
>
> | Method | A (NMSE) | **PA** (NMSE) | AQ (NMSE) | PAQ (NMSE) | A (EMD) | **PA (EMD)** | AQ (EMD) | PAQ (EMD) |
> |-------|----|---------|-----|------|----------|--------------|------------|-------------|
> | **NNOMP** | 0.98 | 0.97 | **0.94** | **0.93** | 29.78 | 27.98 | **24.12** | **23.28** |
> | **WNNOMP** | 48.15 | 23.69 | **0.95** | **0.95** | 23.15 | 23.08 | 24.43 | **23.08** |
> | **ADMM** | 0.88 | 0.86 | **0.81** | __**0.81**__ | 27.13 | 25.97 | **22.28** | __**22.28**__ |
>
> ##### (4, 240, 0), K=10
>
> | Method | A (NMSE) | **PA** (NMSE) | AQ (NMSE) | PAQ (NMSE) | A (EMD) | **PA (EMD)** | AQ (EMD) | PAQ (EMD) |
> |-------|----|---------|-----|------|----------|--------------|------------|-------------|
> | **NNOMP** | 1.00 | 1.00 | **0.98** | **0.98** | 32.94 | 30.01 | **21.63** | **22.10** |
> | **WNNOMP** | 119.49 | 32.65 | **1.01** | **1.01** | 27.46 | 26.31 | **21.61** | __**21.55**__ |
> | **ADMM** | 0.99 | 0.99 | **0.97** | __**0.97**__ | 32.69 | 26.99 | **24.82** | **24.46** |
>
> ##### (6, 141, 1), K=5
>
> | Method | A (NMSE) | **PA** (NMSE) | AQ (NMSE) | PAQ (NMSE) | A (EMD) | **PA (EMD)** | AQ (EMD) | PAQ (EMD) |
> |-------|----|---------|-----|------|----------|--------------|------------|-------------|
> | **NNOMP** | 0.99 | 0.98 | **0.95** | **0.93** | 29.36 | 28.76 | **24.02** | **23.78** |
> | **WNNOMP** | 141.48 | 50.12 | **0.99** | **0.99** | 27.14 | 27.02 | **25.92** | **25.53** |
> | **ADMM** | 0.92 | 0.92 | **0.90** | __**0.90**__ | 22.53 | 20.39 | **15.90** | __**15.90**__ |
>
> ##### (6, 141, 1), K=10
>
> | Method | A (NMSE) | **PA** (NMSE) | AQ (NMSE) | PAQ (NMSE) | A (EMD) | **PA (EMD)** | AQ (EMD) | PAQ (EMD) |
> |-------|----|---------|-----|------|----------|--------------|------------|-------------|
> | **NNOMP** | 1.06 | 1.05 | **1.02** | **1.00** | 29.29 | 27.77 | **21.51** | **21.04** |
> | **WNNOMP** | 157.47 | 49.62 | **1.05** | **1.03** | 31.71 | 28.77 | **20.66** | __**20.34**__ |
> | **ADMM** | 0.92 | 0.90 | **0.79** | __**0.79**__ | 32.95 | 27.69 | **25.55** | **25.55** |

---

> ### Author Response · Authors · 2025-11-24
> **Response to Reviewer qyrv (Part III)**
>
> ## Weakness 1
> We appreciate that the reviewer has accurately captured the core idea of this work. Indeed, there is a trade-off between a better-conditioned sensing matix and an essentially unbiased signal, as you have pointed out. What we emphasize, however, is that such trade-off can be beneficial. Altough $\mathbf{Q}^{-1}\mathbf{x}$ do distort $\mathbf{x}$, it preserves the sparsity and support of $\mathbf{x}$--these are the essential features needed to solve $\mathbf{y} = \mathbf{Ax}$, as once the support is found, the equation can be sovled succinctly via least square (LS).
>
> This assessment is fully supported by our theoretical analysis and empirical results. From the authors’ perspective, the support identification gain obtained by improving the sensing matrix structure outweighs the moderate distortion introduced in the recovered signal, and thus represents a reasonable and practically valuable compromise. Overall, our approach tunes a nearly unsolvable problem to one that permits lossy but informative recovery, which is meaningful.
>
> ## Weakness 2
> In the manuscript, the optimization problem in (3) imposes only two constraints on the preconditioner: invertibility and sparsity preservation. This formulation is justified from two perspectives. First, in sparse signal recovery, particularly in the challenging scenarios involving highly ill-conditioned sensing matrices, the most critical and difficult task is to reliably identify the support of $\mathbf{x}$. In many application domains, accurate support recovery is far more important than precise coefficient estimation. For instance, in gene-based disease classification, the key objective is to determine which genes are associated with the disease, rather than to estimate their exact contribution weights. Likewise, in the LSCM application, correctly identifying the signal propagation angles is essential for downstream tasks such as spectral efficiency prediction. Second, once a reliable support estimate is obtained, the corresponding coefficients can be recalibrated with other specifically designed methods, for example local search or LS-based magnitude correction. With such a post-processing step, the randomness introduced by the proposed stochastic preconditioner can be effectively removed from the final coefficient estimates. We plan to further explore and formalize this coefficient refinement procedure in future work.
>
> ## Weakness 3
> Thanks for the reviewer’s comment regarding the experimental validation of both signal recovery and support recovery. In the manuscript, we evaluate the proposed method on three representative sparse recovery applications: MRI reconstruction, LSCM, and gene-based disease classification. We adopt the most relevant and widely accepted metrics for each task.
>
> For LSCM, we report both NMSE and EMD. NMSE quantifies the overall reconstruction accuracy, while EMD captures the similarity between the estimated and ground truth angular power spectrum by jointly reflecting the support alignment and coefficient distribution, as explained in **Response to Question 6**. This directly addresses the reviewer’s concern about measuring support-level performance.
>
> For MRI reconstruction, we use PSNR to compare image quality across methods. PSNR is the most widely used metric in the compressed-sensing based MRI and other image reconstruction literature [1-5]. We will additionally include the structural similarity index (SSIM) in the revised manuscript to provide a more comprehensive assessment.
>
> For the gene-based leukemia classification task, the ground-truth gene activation pattern, i.e., the true support, is unknown. In such cases, the most meaningful metric is the classifier’s prediction accuracy, which reflects the ultimate task performance [6].
>
> Overall, except for the semi-synthetic LSCM setup where the true support is available, the underlying sparsity pattern is either unknown or not strictly valid. Therefore, we follow standard practice and evaluate performance using downstream task metrics, which appropriately reflect the benefit of the proposed method in each application.

---

> ### Author Response · Authors · 2025-11-24
> **Response to Reviewer qyrv (Part IV)**
>
> ## Weakness 4
> There might be a misunderstanding. We clarify that multiple aspect-ratio configurations have indeed been considered in the manuscript, corresponding to the characteristics and constraints of each application.
>
> In particular, we evaluate five sampling rates ranging from $10\%$ to $50\%$ in the MRI reconstruction experiment. With the image size fixed at $128 \times 128$, i.e., 16384 pixels, these sampling rates result in the following aspect ratios: 1639:16384, 3277:16384, 4916:16384, 6554:16384, and 8192:16384. These settings cover a broad range of measurement regimes and reflect typical sampling rates used in compressed sensing MRI studies.
>
> For the LSCM, the aspect ratio is fixed at 40:6552. This ratio is determined by system-level constraints: the number of available probing beams is specified by the fifth-generation wireless communication new radio standard (8 synchronization signal and physical broadcast channel block beams and 32 channel state information reference signal beams), while the angular domain is discretized into 6552 spatial grids. As a result, the aspect ratio is inherently fixed for this application.
>
> In the gene-based leukemia classification task, the aspect ratio depends on the number of training samples. We evaluate four different training set sizes, i.e., 15, 20, 25, and 30, against 7129 gene features, leading to aspect ratios of 15:7129, 20:7129, 25:7129, and 30:7129. These settings reflect realistic high-dimensional, small-sample regimes commonly encountered in bioinformatics.
>
> ## Weakness 5
> Thanks very much for your valuable comments regarding the extent of the experimental results. We fully agree that a larger-scale evaluation could further strengthen the empirical evidence of our method.
>
> In the initial submission, we selected six representative images from the Kaggle Brain Tumor MRI Dataset, covering pituitary, glioma, meningioma, and no-tumor cases, as well as the commonly used phantom. These images were chosen for two reasons. First, images of the same pathology type typically share similar structural characteristics. Therefore, one or two representative images are sufficient to reflect the general reconstruction behavior for each class. This practice is also common in many other MRI reconstruction studies [1-5]. Second, the current manuscript already evaluates multiple sampling patterns, sampling rates, and recovery algorithms, providing a fairly comprehensive assessment of the proposed method. In addition, we incorporate SSIM as an additional metric in the revised version to further strengthen the evaluation.
>
> To address the reviewer’s concern, we additionally provide experiments on another MRI dataset to further validate the effectiveness of the proposed method. Since the dataset suggested by the reviewer (https://openneuro.org) mainly provides raw MRI measurements that require domain-specific preprocessing beyond our current scope, we instead identify a publicly available dataset containing directly MRI images: the Augmented Alzheimer MRI Dataset on Kaggle available at https://www.kaggle.com/datasets/uraninjo/augmented-alzheimer-mri-dataset. Four additional images from different categories in this dataset is shown in [anonymous link](https://anonymous.4open.science/r/ICLR-2026-Rebuttal-AF33), labeled by 'VeryMildDemented', 'MildDemented', 'ModerateDemented', and 'NoDemented', for reviewer's reference.
>
> The experiments on these MRI images corresponding to four stages of Alzheimer’s disease are summarized in the following tables. Although these images belong to different clinical stages, they exhibit similar anatomical structures and visual characteristics. In this experiment, the proposed method consistently improves upon the conventional approaches. The results lead to conclusions that align with those presented in the main manuscript and further reinforce both the effectiveness and the versatility of the proposed method.
>
> ##### Sampling Rate: 30% (Cartesian)
>
> | Method | Very Mild Demented A | Very Mild Demented AQ | Mild Demented A | Mild Demented AQ |
> |--------|-----------|-------------|------------|-------------|
> | **ISTA PSNR** | 25.82 | **27.25** | 26.09 | **26.62** |
> | **ADMM PSNR** | 26.06 | **27.36** | 26.12 | **26.81** |
> | **ISTA SSIM** | 0.32 | **0.52** | 0.33 | **0.57** |
> | **ADMM SSIM** | 0.32 | **0.54** | 0.34 | **0.57** |
> | **ISTA Aliasing** | 0.10 | **0.09** | 0.10 | **0.08** |
> | **ADMM Aliasing** | 0.10 | **0.08** | 0.10 | **0.08** |
>
> | Method | Moderate Demented A | Moderate Demented AQ | No Demented A | No Demented AQ |
> |--------|-----------|-------------|----------------|----------------|
> | **ISTA PSNR** | 24.42 | **26.17** | 26.22 | **26.45** |
> | **ADMM PSNR** | 24.65 | **26.19** | 26.24 | **26.46** |
> | **ISTA SSIM** | 0.26 | **0.48** | 0.36 | **0.46** |
> | **ADMM SSIM** | 0.28 | **0.49** | 0.37 | **0.46** |
> | **ISTA Aliasing** | 0.07 | 0.07 | 0.08 | **0.07** |
> | **ADMM Aliasing** | 0.07 | 0.07 | 0.08 | **0.07** |

---

> ### Author Response · Authors · 2025-11-24
> **Response to Reviewer qyrv (Part V)**
>
> ## Reference
> [1] Ding, Yanyun, et al, "Efficient dual ADMMs for sparse compressive sensing MRI reconstruction," *Mathematical Methods of Operations Research*, 97.2: 207-231, 2023.
>
> [2] Lyu, Mengye, et al, "M4Raw: A multi-contrast, multi-repetition, multi-channel MRI k-space dataset for low-field MRI research," *Scientific Data*, 10.1: 264, 2023.
>
> [3] Cohen, Regev, Michael Elad, and Peyman Milanfar, "Regularization by denoising via fixed-point projection (RED-PRO)," *SIAM Journal on Imaging Sciences*, 14.3: 1374-1406, 2021.
>
> [4] Ryu, Ernest, et al, "Plug-and-play methods provably converge with properly trained denoisers," *International Conference on Machine Learning (ICML)*, 2019.
>
> [5] Eksioglu, Ender M, "Decoupled algorithm for MRI reconstruction using nonlocal block matching model: BM3D-MRI," *Journal of Mathematical Imaging and Vision*, 56.3: 430-440, 2016.
>
> [6] Zou, Hui, and Trevor Hastie, "Regularization and variable selection via the elastic net," *Journal of the Royal Statistical Society Series B: Statistical Methodology*, 67.2: 301-320, 2005.

---

> > ### Author Response · Authors · 2025-11-27
> > **Follow-up Regarding Discussion on Paper**
> >
> > Dear Reviewer,
> >
> > Hope this message finds you well. As the end of the discussion period is approaching, we would like to kindly follow up to ensure that we have adequately addressed your concerns in our rebuttal. If there are any additional points, clarifications, or feedback you would like us to consider, please feel free to let us know.
> >
> > Your insights are extremely valuable to us, and we would be very glad to further clarify any remaining issues to help improve the paper.
> >
> > Thank you again for your time and effort in reviewing our work.
> >
> > Warm regards,
> >
> > Authors

---

### Official Review · Reviewer_J23z · 2025-10-31

**Soundness:** 3
**Presentation:** 3
**Contribution:** 3
**Rating:** 6
**Confidence:** 3

**Summary:**

The paper tackles sparse recovery problems where the sensing matrix is highly ill-conditioned—characterized by very high aspect ratios ($p\ll n$), heavy-tailed inter-column correlations, and near-proportional columns—conditions that severely violate RIP and render standard compressed sensing (CS) fragile. The authors propose DogRot, a plug-and-play right preconditioner $Q$, applied as $y = A Q Q^{-1} x + n$ with the effective sensing matrix $AQ$. DogRot is constructed as $I$ plus a small-variance, skew-symmetric Gaussian component, making it near-orthogonal and invertible almost surely.

Main claims and results:

Theory: 1) Invertibility (almost sure) of $Q$. 2) Column norm concentration and asymptotic orthogonality of $Q$’s columns, leading to $Q^T Q=(\epsilon (n-1)+1) I$ for large $n$. 3) A concentration bound showing that mutual coherence of $AQ$ contracts relative to that of $A$ by a factor $< 1$, with tail bounds that improve with $n$ and $\epsilon$. 4) Analysis of sparsity preservation: although $Q^{-1}x$ is no longer exactly sparse, hard-thresholding recovers the support with high probability for appropriately chosen variance $\epsilon$ and threshold $t$, trading off decorrelation and amplitude fidelity.

Implementation: A simple algorithm that normalizes $A$, mixes with $Q$, runs any standard sparse recovery routine, and inverts with $Q^{T} $ (as an approximation for $Q^{-1}$) plus hard-thresholding.

Empirics: 1) MRI reconstruction: mutual coherence reductions are substantial (e.g., from $0.99$ to $0.33$ in Cartesian sampling), translating into consistent PSNR gains ($1–3$ dB) for ISTA/ADMM under $20–30$% sampling, both Cartesian and radial. 2) Localized Statistical Channel Modeling (LSCM): On extreme fat matrices (e.g., $40\times 6552$), DogRot markedly reduces mutual coherence and improves APS estimation as measured by NMSE and Earth Mover’s Distance (EMD), with further gains when combined with conventional projectors. 3) Gene-based disease classification: On the $7129$-gene leukemia dataset with very few samples, DogRot reduces MC and improves classification accuracy for LASSO and elastic net, with larger gains in low-sample regimes.

Overall, DogRot is a lightweight, easily integrable preconditioner that consistently reduces mutual coherence and improves support recovery and reconstruction quality across applications.

**Strengths:**

Originality: 1) Reframes preconditioning for CS with highly ill-conditioned matrices by right-multiplying with a randomized, diagonal-dominant near-rotation (as opposed to the more common left-projectors). 2) Introduces a structurally simple but theoretically analyzable class of mixers (skew-symmetric Gaussian perturbations of identity), bridging random matrix intuition with CS identifiability via mutual coherence. 3) Focuses explicitly on the “HIM” regime (heavy-tailed inter-column correlations, extreme aspect ratio), where many standard MC-reduction techniques degrade or fail.

Quality: 1) Provides clear theoretical properties: invertibility, norm concentration, asymptotic orthogonality, and coherence contraction with explicit probabilistic bounds. The analysis acknowledges dependencies and leverages sub-exponential concentration and a Delta-method approximation for normalized inner products. 2) Offers a practical and low-overhead algorithmic pipeline; complexity is dominated by a single matrix multiply with $A$ and an $O(n^2)$ mixing/demixing step. 3) Experiments span diverse domains (MRI, LSCM, genomics), reinforcing generality. The reported MC reductions are large and stable; performance metrics (PSNR, NMSE, EMD, accuracy) consistently improve.

Clarity: 1) The HIM setting is well-motivated and crisply defined (aspect ratio, size, MC, heavy-tail behavior). 2) The trade-off between decorrelation (σ larger) and amplitude fidelity/support preservation (σ smaller) is explained with guidance on threshold selection. 3) Algorithmic steps are succinct; the role of normalization and post-thresholding is explicit.

Significance: 1) Addresses a practical and common pain point: fat, highly correlated sensing matrices in real systems (MRI aliasing, beam similarity in channel estimation, gene expression datasets). 2) The plug-and-play nature means immediate applicability with existing solvers; also composable with existing projectors for additive gains. 3) Potential to expand reliable recovery regimes without redesigning the core solver stack.

**Weaknesses:**

Approximation of $Q^{-1}$ by $Q^T$: 1) Theoretical arguments for using $Q^T$ as an approximation rely on asymptotic orthogonality ($Q^TQ\approx cI$). In finite $n$ and moderate $\epsilon$, the quality of this approximation can vary. The paper would benefit from quantitative finite-sample error bounds on using $Q^T$ instead of the exact $Q^{-1}$, and ablation showing the sensitivity of performance to this approximation.

Parameter selection ($\epsilon$, threshold $t$): While the paper provides qualitative guidance and a conservative threshold based on Gaussian percentiles, there is limited principled methodology for choosing $\epsilon$ across tasks or adapting it to $A$’s statistics (e.g., mutual coherence or spectrum). An automatic, data-driven selection rule (or cross-validation protocol) would improve usability.

Mutual coherence focus: MC is a surrogate for identifiability, but not always fully predictive of recovery performance. The theoretical results do not tie directly to support recovery guarantees for specific algorithms (e.g., OMP/ISTA) under noise, nor to more modern measures (e.g., RIP). Stronger links or bounds (even if conservative) would strengthen the theoretical contribution.

Support recovery claims rely on hard-thresholding: 1) The claim that hard-thresholding recovers support depends on unknown magnitudes and $\epsilon$. The paper does not provide a formal support recovery theorem with explicit SNR or minimum-separation conditions (e.g., $\min |x_j|$ vs. $\epsilon$, $n$, mutual coherence). This is especially relevant under noise and model mismatch.

Experimental scope and reporting details: 1) MRI: Results are promising but mostly reported as PSNR; structural similarity (SSIM) and visual artifact metrics would complement. Sensitivity to sampling masks beyond Cartesian/radial (e.g., Poisson-disc, variable-density) would broaden evidence. 2) LSCM: NMSE values remain high (acknowledged). More analysis disentangling index recovery vs. magnitude recovery would be valuable (e.g., top-$k$ support precision). 3) Genomics: While accuracy improves, additional interpretability checks (e.g., stability of selected genes across seeds, overlap with known biomarkers) would bolster the application impact.

Robustness and randomness: 1) DogRot is stochastic. Although the mutual coherence reduction’s variance is small (~0.01), it would be useful to report performance variance across multiple $Q$ draws, especially for downstream metrics (PSNR, NMSE, accuracy), and discuss whether multiple draws or ensembles can further stabilize results.

**Questions:**

Finite-sample analysis of $Q$ approximation: Can you provide explicit finite-$n$ bounds on the deviation of $Q^T$ from $Q^{-1}$ and quantify how this impacts reconstruction error? For practical $n$ (e.g., 1k–10k), what is the expected condition number of $Q$ and how does it scale with $\epsilon$?

Parameter selection: 1) Could you propose a practical, automatic rule for $\epsilon$ (e.g., function of $\mu(A)$, spectral decay, or empirical Gram statistics)? Would a simple line-search over $\epsilon$ using an objective be effective and cheap? 2) For threshold $t$, beyond the $2.58\sqrt{\epsilon}\|x\|_2$ bound, can you suggest a data-driven surrogate that does not depend on $x$ (e.g., based on noise level estimates or residual statistics)?

Stronger theoretical guarantees beyond mutual coherence:  1) Can you give algorithm-specific (e.g., OMP/ISTA) sufficient conditions after mixing that directly relate to $\epsilon$? 2) Is it possible to characterize how DogRot affects the restricted isometry constants (RIC) or the nullspace property in expectation?

Noise robustness and SNR: How does measurement noise $n$ interact with the mixing and thresholding? Can you provide empirical curves of recovery error vs. SNR with and without DogRot, and guidance on adjusting $\epsilon$ and $t$ under noise?

Randomness and reproducibility: How sensitive are results to the random seed of $Q$? Would averaging reconstructions over a small ensemble of DogRot draws further stabilize or improve results?

Computational overhead in large-scale settings: 1) In extreme $n$ (e.g., $\geq 10^5$), $O(n^2)$ mixing can be heavy. Do you foresee structured DogRot variants (block-diagonal, banded skew-symmetric, or fast transforms) to reduce computational and memory cost?

---

> ### Author Response · Authors · 2025-11-24
> **Response to Reviewer J23z (Part I)**
>
> ## Question 1
> Many thanks for the reviewer's comments about the approximation error by  ${\textstyle{1 \over {\epsilon \left( {n - 1} \right) + 1}}}{\mathbf{Q}}_{\epsilon,n}^T$ in the non-asymptotic regimes and the resulted effect on reconstruction.
>
> The derivation for this deviation and the following influence on recovery is provided in [anonymous link](https://anonymous.4open.science/r/ICLR-2026-Rebuttal-AF33) from Figure '01.png' to '05.png'. The deviation and condition number will keep a low level with proper parameter setting. We also explain that this approximation would not violate signal property and propagate additional error for reconstruction. We will supplement the discussion in the revised manuscript.
>
> ## Question 2
> Thanks for the reviewer's comment on a more practical and automatic rule to select the hyper-parameters involved in the proposed algorithm. As analyzed in the manuscript, $\epsilon$ serves as an extremely important role in balancing the reduction of mutual coherence and signal recoverability. The hard-thresholding parameter $t$ is also critical in support recognition for those applications the cardinality of the signal of interest $\mathbf{x}$ is unknown.
>
> Detailed derivations and discussion with repsect to the selection criterion for the parameters are provided in [anonymous link](https://anonymous.4open.science/r/ICLR-2026-Rebuttal-AF33) in Figure '06.png' to '07.png'.
>
> ## Question 3
> Thanks for the reviewer's comment on stronger theoretical guarantees beyond mutual coherence.
>
> The detailed theoretical discussion about sufficient condition for OMP and ISTA, and the null-space property improved by the proposed DogRot is also provided in [anonymous link](https://anonymous.4open.science/r/ICLR-2026-Rebuttal-AF33) from Figure '08.png' to '13.png'.
>
> ## Question 4
> Thanks for the reviewer’s comment on noise robustness and SNR. In fact, the measurement noise is independent of the proposed mixing operation, whose purpose is to improve the properties of the sensing matrix. The noise only affects the performance of the recovery algorithm itself. Consequently, the conclusions established in the existing literature can be directly applied to our method. The relevant results for OMP and ISTA, along with guidelines for adjusting $\epsilon$ and $t$ in the presence of noise, have also been provided in the above discussion. Additionally, the empirical curves of recovery error vs. SNR with and without DogRot exampled by the LSCM and relavant discussion are simultaneously provided in [anonymous link](https://anonymous.4open.science/r/ICLR-2026-Rebuttal-AF33) from Figure '08.png' to '13.png'.

---

> ### Author Response · Authors · 2025-11-24
> **Response to Reviewer J23z (Part II)**
>
> ## Question 5
> We thank the reviewer for raising the question regarding the robustness of the proposed DogRot with respect to random seeds. All experimental results reported in the manuscript were averaged over multiple independent Monte-Carlo experiments. To further assess the stability across different realizations of $\mathbf{Q}_{\epsilon,n}$ we additionally report the variance of the key results. Specifically, for each application scenario considered in this work, we select one representative experimental setting and record both the resulting mutual coherence and one corresponding performance metric across multiple independent draws. The results are also provided in [anonymous link](https://anonymous.4open.science/r/ICLR-2026-Rebuttal-AF33). Please check Figure 'MC_MRI', 'MC_LSCM', 'MC_Gene', 'PSNR_MRI', 'mri_image5', 'NMSE_LSCM', 'NMSE_LSCM_spec', 'Acc_Gene', and 'Acc_Gene_spec'.
>
> In these figures, we report the mutual coherence and one corresponding performance metric achieved by both the conventional method and the proposed DogRot across 1,000 independent experiments. A representative setting is selected for each application. The mutual coherence comparison is illustrated in Figure 'MC_MRI', 'MC_LSCM', and 'MC_Gene' using a box plot, where the dashed line indicates the mutual coherence of the original sensing matrix. As expected, the results vary across experiments due to the stochastic nature of the proposed method. Nevertheless, the resulting mutual coherence consistently remains below the baseline and exhibits only moderate variability.
>
> The PSNR results over 1,000 independent experiments for reconstructing a meningioma image using ISTA are shown in Figure 'PSNR_MRI'. 20\% Cartesian sampling and Figure 'mri_image5' is adopted. Similar to the mutual coherence, the proposed method exhibits moderate variability in the achieved PSNR values while consistently outperforming the conventional method.
>
> The NMSE results across 1,000 independent experiments for LSCM using NNOMP are shown in Figures 'NMSE_LSCM' and 'NMSE_LSCM_spec'. In Figure 'NMSE_LSCM', the ground truth signal $\mathbf{x}$ is randomly generated in each experiment. It can be observed that both methods exhibit pronounced fluctuations in NMSE due to the highly ill-conditioned nature of the problem, where successful recovery is strongly influenced by the specific realization of $\mathbf{x}$. Nevertheless, it is evident that, on average, the proposed method achieves lower NMSE than the conventional approach. In Figure 'NMSE_LSCM_spec', independent experiments are conducted with a fixed $\mathbf{x}$ and randomly drawn $\mathbf{Q_{\epsilon, n}}$. In this scenario, the proposed method still outperforms the conventional one, although the performance gains are not always substantial. However, this observation highlights the importance of the reviewer’s suggestion regarding enhancing robustness through average.
>
> Figures 'Acc_Gene' and 'Acc_Gene_spec' report the classification accuracy over 1,000 independent experiments for the leukemia dataset using $\ell_1$-regularized logistic regression, where 30 samples are used for training. In Figure 'Acc_Gene', the training samples are randomly selected for each run, whereas Figure 'Acc_Gene_spec' presents the results obtained with a fixed training set. Similar to the previous experiment, the proposed method statistically and consistently outperforms the conventional approach. However, in certain extreme cases, the proposed method may yield slightly worse performance than the baseline. These observations further support the reviewer's suggestion that, in some applications, averaging reconstructions over a small ensemble of DogRot realizations may help stabilize and potentially improve performance. Implementing and analyzing this ensemble-based strategy requires additional theoretical justification and empirical investigation, which we plan to pursue in future work.

---

> ### Author Response · Authors · 2025-11-24
> **Response to Reviewer J23z (Part III)**
>
> ## Question 6
> We thank the reviewer for raising the question regarding the computational complexity of the proposed DogRot when $n$ is large. Compared with existing reconstruction algorithms, DogRot introduces only two additional operations: mixing the sensing matrix and detransforming the recovered signal. These steps incur computational complexities of $\mathcal{O}(pn^2)$ and $\mathcal{O}(n^2)$, respectively. For reference, the complexities of standard reconstruction algorithms such as vanilla OMP, ADMM, and ISTA are approximately $\mathcal{O}(Tpn)$, where $T$ denotes the number of iterations. Therefore, when $n$ becomes large, DogRot may require noticeably higher computational cost than conventional methods. This increased complexity is the tradeoff for achieving improved reconstruction performance.
>
> At the current stage, by exploiting the skew-symmetric structure of DogRot, the cost of the matrix mixing step can be reduced to $\mathcal{O} ({\textstyle{1 \over 2}}pn^2)$. Using advanced computational tools, such as GPU, can also accelerate the processing. However, this does not change the complexity order in terms of $n$. In this regard, the structured DogRot variants suggested by the reviewer, e.g., block-diagonal, banded skew-symmetric, or fast transforms-based constructions, are indeed promising for further reducing computation and memory overhead. Their impact on mutual coherence reduction and the preservation of the theoretical properties of DogRot remains an open question and warrants further investigation. We appreciate these constructive suggestions, which will guide and enhance our future research.
>
> ## Weakness 1
> Please see **Response to Question 1**.
>
> ## Weakness 2
> Please see **Response to Question 2**.
>
> ## Weakness 3 & Weakness 4
> The responses to these two issues rely on similar reasoning and are supported by the same evidence and materials provided in **Response to Question 3 & Question 4**.

---

> ### Author Response · Authors · 2025-11-24
> **Response to Reviewer J23z (Part IV)**
>
> ## Weakness 5
> Thanks for the revirewer's comments on experimental scope and reporting details.
> ### MRI
> To strengthen the empirical evidence, we expand our evaluation by adding SSIM and the aliasing index (as suggested) and by including variable-density Cartesian sampling to assess mask sensitivity.
>
> Regarding Poisson-disc sampling, although recommended, it is a pseudo-random pattern with strong decorrelation, producing sensing matrices with very low mutual coherence (often around 0.2), outside the highly ill-conditioned regime targeted in this work. Moreover, its non-continuous trajectories are impractical for real MRI scanners and mainly used for academic analysis. Thus, we do not include this pattern.
>
> We compare the conventional method with the proposed DogRot across two sampling rate (20\% and 30\%), three sampling patterns (Cartesian, radial, and variable-density Cartesian), six types of brain-tumor images, and three evaluation metrics (PSNR, SSIM, and Aliasing index). Among these configurations, the variable-density Cartesian sampling pattern, together with the SSIM and aliasing-index metrics, are newly added results. The SSIM is expected to be larger for a better reconstruction while the aliasing index should be smaller. The complete set of results is provided in the following.
>
> #### Sampling Rate: 20% (Cartesian)
>
> | Method | Phantom A | Phantom AQ | Notumor A | Notumor AQ | Pituitary A | Pituitary AQ |
> |--------|-----------|-------------|------------|-------------|-------------|---------------|
> | **ISTA PSNR** | 15.03 | **16.08** | 11.71 | **12.95** | 11.82 | **13.30** |
> | **ADMM PSNR** | 15.03 | **16.14** | 11.81 | **13.06** | 12.56 | **13.50** |
> | **ISTA SSIM** | 0.24 | **0.55** | 0.22 | **0.55** | 0.25 | **0.51** |
> | **ADMM SSIM** | 0.25 | **0.55** | 0.23 | **0.56** | 0.25 | **0.51** |
> | **ISTA Aliasing** | 0.22 | **0.18** | 0.16 | **0.12** | 0.13 | **0.10** |
> | **ADMM Aliasing** | 0.22 | **0.17** | 0.15 | **0.12** | 0.13 | **0.10** |
>
> | Method | Glioma A | Glioma AQ | Meningioma1 A | Meningioma1 AQ | Meningioma2 A | Meningioma2 AQ |
> |--------|-----------|-------------|----------------|-----------------|----------------|-----------------|
> | **ISTA PSNR** | 14.91 | **16.04** | 12.51 | **13.99** | 14.34 | **15.98** |
> | **ADMM PSNR** | 14.91 | **16.12** | 12.60 | **14.12** | 14.37 | 13.50 |
> | **ISTA SSIM** | 0.26 | **0.51** | 0.30 | **0.35** | 0.41 | **0.57** |
> | **ADMM SSIM** | 0.27 | **0.51** | 0.30 | **0.36** | 0.41 | **0.57** |
> | **ISTA Aliasing** | 0.15 | **0.12** | 0.11 | **0.09** | 0.07 | **0.06** |
> | **ADMM Aliasing** | 0.15 | **0.12** | 0.11 | **0.09** | 0.07 | **0.06** |
>
> #### Sampling Rate: 20% (Radial)
>
> | Method | Phantom A | Phantom AQ | Notumor A | Notumor AQ | Pituitary A | Pituitary AQ |
> |--------|-----------|-------------|------------|-------------|-------------|---------------|
> | **ISTA PSNR** | 19.71 | **20.79** | 20.53 | **22.15** | 22.77 | **23.60** |
> | **ADMM PSNR** | 19.71 | **20.79** | 20.55 | **22.15** | 22.78 | **23.62** |
> | **ISTA SSIM** | 0.27 | **0.44** | 0.28 | **0.42** | 0.42 | **0.44** |
> | **ADMM SSIM** | 0.27 | **0.46** | 0.28 | **0.42** | 0.43 | **0.46** |
> | **ISTA Aliasing** | 0.22 | **0.21** | 0.13 | **0.12** | 0.12 | 0.12 |
> | **ADMM Aliasing** | 0.21 | **0.20** | 0.12 | 0.12 | 0.11 | 0.11 |
>
> | Method | Glioma A | Glioma AQ | Meningioma1 A | Meningioma1 AQ | Meningioma2 A | Meningioma2 AQ |
> |--------|-----------|-------------|----------------|-----------------|----------------|-----------------|
> | **ISTA PSNR** | 24.36 | **25.22** | 22.95 | **24.88** | 25.19 | **26.11** |
> | **ADMM PSNR** | 24.39 | **25.27** | 22.95 | **24.94** | 26.44 | **27.22** |
> | **ISTA SSIM** | 0.47 | **0.51** | 0.50 | **0.53** | 0.56 | **0.59** |
> | **ADMM SSIM** | 0.47 | **0.51** | 0.50 | **0.54** | 0.56 | **0.60** |
> | **ISTA Aliasing** | 0.14 | 0.14 | 0.09 | 0.09 | 0.07 | 0.07 |
> | **ADMM Aliasing** | 0.14 | **0.13** | 0.09 | 0.09 | 0.07 | 0.07 |

---

> ### Author Response · Authors · 2025-11-24
> **Response to Reviewer J23z (Part V)**
>
> #### Sampling Rate: 20% (Variable-Density Cartesian)
>
> | Method | Phantom A | Phantom AQ | Notumor A | Notumor AQ | Pituitary A | Pituitary AQ |
> |--------|-----------|-------------|------------|-------------|-------------|---------------|
> | **ISTA PSNR** | 20.50 | **21.39** | 20.95 | **22.23** | 23.01 | **23.98** |
> | **ADMM PSNR** | 20.62 | **21.52** | 21.01 | **22.29** | 23.16 | **24.10** |
> | **ISTA SSIM** | 0.36 | **0.42** | 0.36 | **0.50** | 0.49 | **0.59** |
> | **ADMM SSIM** | 0.37 | **0.42** | 0.37 | **0.51** | 0.50 | **0.59** |
> | **ISTA Aliasing** | 0.22 | **0.20** | 0.12 | 0.12 | 0.10 | 0.10 |
> | **ADMM Aliasing** | 0.22 | **0.19** | 0.12 | 0.12 | 0.10 | 0.10 |
>
> | Method | Glioma A | Glioma AQ | Meningioma1 A | Meningioma1 AQ | Meningioma2 A | Meningioma2 AQ |
> |--------|-----------|-------------|----------------|-----------------|----------------|-----------------|
> | **ISTA PSNR** | 24.81 | **26.39** | 22.27 | **24.00** | 27.76 | **27.79** |
> | **ADMM PSNR** | 24.91 | **26.42** | 22.30 | **24.06** | 27.77 | **27.81** |
> | **ISTA SSIM** | 0.53 | **0.66** | 0.61 | **0.66** | 0.69 | **0.72** |
> | **ADMM SSIM** | 0.53 | **0.66** | 0.30 | **0.66** | 0.70 | **0.72** |
> | **ISTA Aliasing** | 0.13 | **0.12** | 0.09 | **0.08** | 0.07 | **0.06** |
> | **ADMM Aliasing** | 0.13 | **0.12** | 0.09 | **0.08** | 0.07 | **0.06** |
>
> #### Sampling Rate: 30% (Cartesian)
>
> | Method | Phantom A | Phantom AQ | Notumor A | Notumor AQ | Pituitary A | Pituitary AQ |
> |--------|-----------|-------------|------------|-------------|-------------|---------------|
> | **ISTA PSNR** | 16.26 | **18.19** | 12.31 | **14.62** | 12.17 | **14.79** |
> | **ADMM PSNR** | 16.26 | **18.19** | 12.50 | **14.77** | 13.10 | **14.79** |
> | **ISTA SSIM** | 0.34 | **0.74** | 0.30 | **0.67** | 0.32 | **0.60** |
> | **ADMM SSIM** | 0.36 | **0.74** | 0.31 | **0.67** | 0.33 | **0.62** |
> | **ISTA Aliasing** | 0.19 | **0.12** | 0.11 | **0.08** | 0.10 | **0.07** |
> | **ADMM Aliasing** | 0.18 | **0.12** | 0.11 | **0.08** | 0.09 | **0.07** |
>
> | Method | Glioma A | Glioma AQ | Meningioma1 A | Meningioma1 AQ | Meningioma2 A | Meningioma2 AQ |
> |--------|-----------|-------------|----------------|-----------------|----------------|-----------------|
> | **ISTA PSNR** | 15.29 | **17.16** | 13.12 | **15.05** | 15.06 | **16.98** |
> | **ADMM PSNR** | 15.29 | **17.17** | 13.21 | **15.14** | 15.08 | **17.05** |
> | **ISTA SSIM** | 0.34 | **0.58** | 0.39 | **0.43** | 0.45 | **0.63** |
> | **ADMM SSIM** | 0.35 | **0.59** | 0.39 | **0.44** | 0.45 | **0.64** |
> | **ISTA Aliasing** | 0.11 | **0.09** | 0.09 | **0.07** | 0.07 | **0.05** |
> | **ADMM Aliasing** | 0.11 | **0.08** | 0.09 | **0.06** | 0.07 | **0.05** |
>
> #### Sampling Rate: 30% (Radial)
>
> | Method | Phantom A | Phantom AQ | Notumor A | Notumor AQ | Pituitary A | Pituitary AQ |
> |--------|-----------|-------------|------------|-------------|-------------|---------------|
> | **ISTA PSNR** | 21.40 | **22.39** | 22.08 | **24.12** | 24.51 | **25.19** |
> | **ADMM PSNR** | 21.40 | **22.39** | 22.15 | **24.12** | 24.60 | **25.22** |
> | **ISTA SSIM** | 0.31 | **0.49** | 0.30 | **0.47** | 0.48 | **0.70** |
> | **ADMM SSIM** | 0.32 | **0.50** | 0.30 | **0.49** | 0.48 | **0.70** |
> | **ISTA Aliasing** | 0.20 | **0.19** | 0.12 | 0.12 | 0.11 | **0.10** |
> | **ADMM Aliasing** | 0.19 | 0.19 | 0.12 | 0.12 | 0.10 | 0.10 |
>
> | Method | Glioma A | Glioma AQ | Meningioma1 A | Meningioma1 AQ | Meningioma2 A | Meningioma2 AQ |
> |--------|-----------|-------------|----------------|-----------------|----------------|-----------------|
> | **ISTA PSNR** | 26.26 | **27.22** | 25.17 | **26.09** | 28.32 | **29.39** |
> | **ADMM PSNR** | 26.26 | **27.25** | 25.21 | **26.12** | 28.92 | **30.42** |
> | **ISTA SSIM** | 0.53 | **0.71** | 0.61 | **0.63** | 0.57 | **0.65** |
> | **ADMM SSIM** | 0.53 | **0.72** | 0.62 | **0.63** | 0.58 | **0.66** |
> | **ISTA Aliasing** | 0.13 | **0.12** | 0.09 | 0.09 | 0.07 | 0.07 |
> | **ADMM Aliasing** | 0.12 | 0.12 | 0.09 | **0.08** | 0.06 | 0.06 |

---

> ### Author Response · Authors · 2025-11-24
> **Response to Reviewer J23z (Part VI)**
>
> #### Sampling Rate: 30% (Variable-Density Cartesian)
>
> | Method | Phantom A | Phantom AQ | Notumor A | Notumor AQ | Pituitary A | Pituitary AQ |
> |--------|-----------|-------------|------------|-------------|-------------|---------------|
> | **ISTA PSNR** | 22.80 | **24.16** | 26.80 | **27.35** | 23.99 | **27.67** |
> | **ADMM PSNR** | 22.91 | **24.24** | 26.84 | **27.36** | 24.16 | **27.76** |
> | **ISTA SSIM** | 0.37 | **0.45** | 0.54 | **0.61** | 0.53 | **0.69** |
> | **ADMM SSIM** | 0.37 | **0.45** | 0.55 | **0.61** | 0.54 | **0.70** |
> | **ISTA Aliasing** | 0.18 | **0.15** | 0.10 | 0.10 | 0.09 | **0.08** |
> | **ADMM Aliasing** | 0.17 | **0.15** | 0.10 | 0.10 | 0.08 | **0.07** |
>
> | Method | Glioma A | Glioma AQ | Meningioma1 A | Meningioma1 AQ | Meningioma2 A | Meningioma2 AQ |
> |--------|-----------|-------------|----------------|-----------------|----------------|-----------------|
> | **ISTA PSNR** | 28.01 | **28.55** | 26.30 | **28.17** | 29.91 | **32.22** |
> | **ADMM PSNR** | 28.21 | **28.58** | 26.63 | **28.31** | 29.99 | **32.36** |
> | **ISTA SSIM** | 0.65 | **0.71** | 0.73 | **0.76** | 0.73 | **0.82** |
> | **ADMM SSIM** | 0.66 | **0.71** | 0.73 | **0.77** | 0.73 | **0.82** |
> | **ISTA Aliasing** | 0.11 | 0.11 | 0.11 | **0.07** | 0.06 | **0.05** |
> | **ADMM Aliasing** | 0.11 | **0.10** | 0.11 | **0.06** | 0.05 | 0.05 |
>
> These results reveal that the proposed DogRot generally outperforms the conventional method across different sampling patterns, sampling rates, and image types under all evaluation criteria. In particular, for Cartesian and variable-density Cartesian sampling, where aliasing artifacts are more likely to occur, the performance gain introduced by DogRot is especially significant. Since the results obtained under variable-density Cartesian sampling and the aliasing index provide only marginal additional insights and do not lead to further meaningful conclusions, we will include only the complementary SSIM results in the revised manuscript to strengthen the evidence while maintaining conciseness.
>
> ### LSCM
> In our manuscript, we provided NMSE and EMD results achieved by different algorithms under different base station configurations. The NMSE is a metric evaluating the overall reconstruction performance. Figure 'APS' in [anonymous link](https://anonymous.4open.science/r/ICLR-2026-Rebuttal-AF33) would help the reviewer to better understand the LSCM problem. This figure explains why and how both the conventional and proposed methods performs poorly in terms of NMSE.
>
> Although the NMSE remains large, the proposed method produces APS estimates visibly closer to ground truth, turning an almost unsolvable problem into an approximate but informative reconstruction. However, with only 1 of 5 paths recovered exactly, support-based metrics such as top-$k$ support precision capture at most a 20\% gain. Moreover, if the estimated angles all fall near, but not exactly on, the true indices, the top-$k$ support precision would report no improvement at all, even though the reconstruction is meaningfully better. Thus, such index-only metrics fail to capture the full advantage of our method.
>
> For this reason, we adopt EMD to quantify performance. EMD measures the minimum transport cost between two distributions, accounting for both support alignment and magnitude similarity. This optimal-transport formulation effectively captures the distributional improvements achieved by our method, which neither NMSE nor top-$k$ support precision can reflect in this highly ill-conditioned LSCM setting. Therefore, we believe EMD is the most suitable metric for evaluating APS recovery performance in our scenario.

---

> ### Author Response · Authors · 2025-11-24
> **Response to Reviewer J23z (Part VII)**
>
> ### Genomics
> Thank you for pointing out the importance of interpretability in genomics applications. We fully agree that evaluating the stability and biological relevance of the selected genes can strengthen the impact of our study.
>
> We perform an additional analysis where the model was trained with 1000 different random seeds. For each run, we recorded the selected gene set and quantified stability following the reviewer's suggestion. We adopt the Jaccard similarity to measure the stability of gene selection. Denoting the set of selected gene in the $i$-th experiment as $\mathcal{G_i}$, the Jaccard similarity is calculated by
> $$
> S = \mathbb{E} \frac{\left| \mathcal{G_i}\cap \mathcal{G_j} \right|}{\left| \mathcal{G_i}\cup \mathcal{G_j} \right|}, \forall i \neq j.
> $$
>
> The following table reports the Jaccard similarity achieved by both LASSO and Elastic Net for different numbers of training samples, and compares these results with those obtained using the proposed preconditioning method. The results indicate that both LASSO and Elastic Net exhibit low Jaccard similarity, even with the aid of the proposed method. This behavior is due to the ill-conditioned nature of the problem, where the number of samples is much smaller than the number of genes. LASSO shows the lowest stability, as the number of genes it can select is limited to at most the number of samples, and the proposed method cannot improve this. In contrast, Elastic Net allows more genes to be selected, so the proposed method can provide greater improvement in stability. Overall, these results indicate that the proposed method achieves stability that is at least comparable to that of the conventional method.
>
> #### Jaccard Similarity
>
> | Sample Number | 15 | 20 | 25 | 30 |
> |-------|----|---------|-----|------|
> | $\ell_1$ regularization A | 0.02 | 0.04 | 0.06 | 0.08 |
> | Elastic Net A | 0.03 | 0.06 | 0.08 | 0.11 |
> | $\ell_1$ regularization AQ | 0.02 | 0.03 | 0.06 | 0.08 |
> | Elastic Net AQ | 0.04 | 0.06 | 0.09 | 0.13 |
>
> Regarding the second point on additional interpretability checks, i.e., overlap with known biomarkers, we fully agree that such analyses would further strengthen the biological insight of the method. However, as our work focuses primarily on the signal processing methodology rather than biological validation, these analyses fall outside the scope and expertise of our current study. We therefore leave this direction for future collaboration with domain experts. Nevertheless, in accordance with the reviewer’s first suggestion, the Jaccard similarity result will strengthen the evaluation of model stability. These results further confirm the robustness and effectiveness of our method.
>
> ## Weakness 6
> Please see **Response to Question 5**.

---

> > ### Author Response · Authors · 2025-11-27
> > **Follow-up Regarding Discussion on Paper**
> >
> > Dear Reviewer,
> >
> > Hope this message finds you well. As the end of the discussion period is approaching, we would like to kindly follow up to ensure that we have adequately addressed your concerns in our rebuttal. If there are any additional points, clarifications, or feedback you would like us to consider, please feel free to let us know.
> >
> > Your insights are extremely valuable to us, and we would be very glad to further clarify any remaining issues to help improve the paper.
> >
> > Thank you again for your time and effort in reviewing our work.
> >
> > Warm regards,
> >
> > Authors

---

### Official Review · Reviewer_VhLj · 2025-11-03

**Soundness:** 3
**Presentation:** 3
**Contribution:** 3
**Rating:** 6
**Confidence:** 3

**Summary:**

A random precondition is introduced to deal with the high coherence in the measurement matrix for compressed sensing, and detailed analysis is provided.

**Strengths:**

New random scheme for reduce the mutual coherence of a CS matrix.

**Weaknesses:**

Overall an interesting paper.

**Questions:**

* Any more intuition behind the construction of $Q_{\epsilon,n}$?
* I didn't quite get whether exact recovery is still possible after the precondition?

---

> ### Author Response · Authors · 2025-11-24
> **Response to Reviewer VhLj**
>
> Thanks for your overall positive recommendation. We would like to further discuss your two problems together as they are closely related. (Some equations are not arranged strictly since they cannot be properly compiled)
>
> The intuition behind the construction of $Q_{\epsilon,n}$ is simple: The main reason that hinders the identifiability of $\mathbf{y} = \mathbf{A} \mathbf{x}$ is the $\mathbf{A}$ contains many nearly proportional columns (referred to as ill-conditioned). Hence, our idea is to disentangle them by applying random perturbations--matrix $Q_{\epsilon,n}$ is the trigger. It combines each column with randomly scaled other columns (possibly less correlated), and thereby the coherence is expected to be reduced. The skew-symmetric strucutre of $\mathbf{Q}$ magically suffices our goal, which makes the columns in $\mathbf{AQ}$ more scattered. An intuitive example is: $ \mathbf{A} $=
> | 1.0 | 0.95 | -1.2 | 0.5 | -0.8 |
> |--------|-----------|-------------|------------|-------------|
> | 0.5 | 0.48 | 1.5 | -1.2 | -1.5 |
>
> where $a_1$ and $a_2$ (the first and second column of $\mathbf{A}$ are nearly proportional. Applying $Q_{\epsilon,n}$ as
> | 1.0 | 0 | -0.6 | 0.1 | 0.1 |
> |--------|-----------|-------------|------------|-------------|
> | 0 | 1 | 0.6 | 0.1 | 0.1 |
> | -0.6  | 0.6 | 1 | 0.1 | 0.1 |
> | -0.1 | -0.1 | -0.1 | 1 | 0.1 |
> | -0.1 | -0.1 | -0.1 | -0.1 | 1 |
>
> also shown in Figure 'example_Q.png' in [anonymous link](https://anonymous.4open.science/r/ICLR-2026-Rebuttal-AF33),
> which tunes $AQ_{\epsilon,n}$ into:
> | $\mathbf{0.31}$ | $\mathbf{1.61}$ | -1.25 | 0.42 | -0.86 |
> |--------|-----------|-------------|------------|-------------|
> | $\mathbf{1.29}$ | $\mathbf{-0.58}$ | 1.35 | -1.40 | -1.56 |
>
> See a visualization in Figure 'example.png' in [anonymous link](https://anonymous.4open.science/r/ICLR-2026-Rebuttal-AF33). This intuitively explain the mechanism of $\mathbf{Q}_{\epsilon,n}$.
>
> When designing $\mathbf{Q}$, we also make sure that it must be diagonally dominant so that the main direction of each column in $\mathbf{A}$ is preserved and the support information in the original signal is not destroyed. The skew-symmetric structure is introduced to constrain the randomness and ensure convenient mathematical properties that facilitate both analysis and implementation.
>
> More intuitions can be found in Figure 'APS.png' in [anonymous link](https://anonymous.4open.science/r/ICLR-2026-Rebuttal-AF33), which compares the ground-truth APS with the estimates obtained by different methods. The conventional approach is unable to recover the underlying structure at all. Even the classical left-multiplying preconditioner ($\mathbf{PA}$) yields only marginal improvement, as confirmed by the analysis in the manuscript.
>
> This design shows promising behavior in simulations, which led us to seek theoretical justification. The theoretical results in the manuscript confirm that DogRot reduces mutual coherence and that the recovered signal corresponds to the original signal perturbed by a controlled Gaussian noise term. The empirical results, especially the APS comparison in the reference figure, demonstrate the same trend, i.e., although exact recovery remains impossible in this extremely challenging LSCM setting, DogRot consistently produces APS estimates that are far closer to the ground truth than those obtained by conventional methods. In other words, DogRot enables a meaningful transition from an almost unsolvable problem to a lossy but informative reconstruction.
>
> In such ill-conditioned regimes, an approximate but structurally correct recovery is substantially more valuable than a completely invalid one. Moreover, once DogRot provides a coarse localization of the signal support, existing refinement techniques, e.g., local search or LS-based magnitude correction, can further improve accuracy. Therefore, we believe DogRot offers an important step toward solving highly ill-conditioned sparse recovery problems by bridging the gap between total failure and practically useful reconstruction.

---

> > ### Author Response · Authors · 2025-11-27
> > **Follow-up Regarding Discussion on Paper**
> >
> > Dear Reviewer,
> >
> > Hope this message finds you well. As the end of the discussion period is approaching, we would like to kindly follow up to ensure that we have adequately addressed your concerns in our rebuttal. If there are any additional points, clarifications, or feedback you would like us to consider, please feel free to let us know.
> >
> > Your insights are extremely valuable to us, and we would be very glad to further clarify any remaining issues to help improve the paper.
> >
> > Thank you again for your time and effort in reviewing our work.
> >
> > Warm regards,
> >
> > Authors

---

### Author Response · Authors · 2025-11-30
**Global Response to Area Chair**

We would like to express our sincere gratitude to all reviewers for their valuable comments and to the area chair for handling our submission. To help the area chair better grasp the discussion surrounding this work, we summarize the reviewers’ key concerns and our corresponding responses below.

Among the three reviewers, VhLj and J23z provide generally positive feedback and ratings, and their comments mainly focus on suggestions for further improving the techinical content of the work. Reviewer qyrv assigns a lower rating, but the concerns raised are relatively limited in scope and focus primarily on typos and clarifications regarding certain experimental settings.

Consequently, in general, **the reviewers acknowledge the novelty, contributions, and broad applicability of our work**. The remaining concerns fall primarily into three categories: clearer representation, stronger theoretical guarantees, and richer experimental evidence. The detailed responses are provided in the following sections.
## Clearer Representation (Reviewer VhLj and qyrv)
### 1. Providing more intuition behind the algorithm’s design and validity (Reviewer VhLj)
Reviewer VhLj suggested adding more intuitions to clarify the design rationale and validity of the proposed algorithm. We have incorporated these explanations and illustrative examples in the response to Reviewer VhLj, which we believe significantly improve the accessibility and clarity of the core ideas and contributions.

### 2. Typos and unclear equation definitions (Reviewer qyrv)
All typos and unclear or inconspicuous definitions have been carefully corrected in the revised manuscript.

All concerns related to representation have been fully addressed.

## Stronger theoretical guarantees (Reviewer J23z and qyrv)
### 1. Justification for the orthogonal approximation (Reviewer J23z and qyrv)
The relevant theoretical justifications are provided in [anonymous link](https://anonymous.4open.science/r/ICLR-2026-Rebuttal-AF33) from Figure '01.png' to '05.png'.

### 2. More practical strategies for determining hyperparameters (Reviewer J23z and qyrv)
Additional theoretical support and practical guidelines are also provided in [anonymous link](https://anonymous.4open.science/r/ICLR-2026-Rebuttal-AF33) in Figure '06.png' and '07.png'.

### 3. Stronger guarantees beyond mutual coherence and algorithm-specific recovery conditions under noise (Reviewer J23z)
We supplement the improvement in the null-space property introduced by our method, and provide recovery conditions for both OMP and ISTA, along with practical hyperparameter selection strategies that explicitly account for SNR. These details appear in [anonymous link](https://anonymous.4open.science/r/ICLR-2026-Rebuttal-AF33) from Figure '08.png' to '13.png'.

All concerns regarding additional theoretical guarantees have been thoroughly addressed.

## Richer experimental evidence (Reviewer J23z and qyrv)
### 1. Justification for experimental settings (Reviewer J23z and qyrv)
The primary concern relates to the use of Earth Mover’s Distance (EMD) in the localized statistical channel modelling (LSCM) experiment. This has been clarified in our individual responses. We further explain why the conventional left-multiplying preconditioning method is not included as a baseline.

### 2. More experiments under diverse scenarios or datasets (Reviewer J23z and qyrv)
We provide extensive new experimental evidence across all three application domains:

1) MRI Reconstruction: The results on an another dataset, sampling pattern and evaluation metric are provided.

2) LSCM: The performance of conventional left-multiplying preconditioning method is provided.

3) Gene-based disease classification: Additional results on Jaccard similarity to evaluate the stability of the proposed method is provided.

Moreover, beyond the average performance across Monte Carlo independent experiments, we also provide box plots to illustrate the stability of the proposed method across individual runs. In summary, all reasonable requests for additional experiments have been satisfied.

## Final Remarks
In conclusion, **since the reviewers generally recognize the novelty and contributions of this work, and all remaining concerns regarding representation, theoretical rigor, and experimental validation have been thoroughly addressed**, we hope that this summary helps the area chair better assess the current status and completeness of the submission. We sincerely appreciate the area chair’s efforts.

---

### Meta-Review · Area_Chair_Tcb5 · 2026-01-05

**Summary:**

The authors propose a simple strategy to deal with an issue in the sparse recovery literature: sensing matrices with bad coherence parameter. The suggestion is to use a family of random matrices to improve coherence, while (hopefully) not blowing up the effective sparsity by too much. The authors acknowledge that tuning must be done to control the tradeoff in effective sparsity. Experiments show that for some parameter choices, coherence of sensing matrix vastly improves while reconstruction improves slightly.

The reviews were warm but not overwhelmingly positive, and I agree with the reviewers concerns that while the improvements to coherence are good, comparisons in terms of SNR or other actual reconstruction metrics seem less convincing. Ultimately, the downstream goal of sparse recovery is recovery, so a strong improvement even in some restricted settings would be the best sell of the method. The lack of such a more compelling application, even with a highly-coherent starting matrix, leads me to recommend rejection.

**Reviewer Concerns:**

I appreciate the authors' engagement with the reviewers' concerns, but I didn't find the new reconstruction results a stark enough qualitative change to tip the scales on the paper's outlook. I think the reviewers would feel similarly. I think the thing that would strengthen the paper most is an experiment where using DogRot leads to consistent strong improvements in reconstruction over the baseline, along with some theoretical motivation (properties of the design matrix / instance that led to such an improvement).

**Reviewer Scores:**

I could see the scores increasingly slightly, but perhaps not enough to cross the threshold.

---

### Decision · Program_Chairs · 2026-01-26

Reject